# Normalizing flow neural networks by JKO scheme

**Chen Xu**
School of Industrial and
Systems Engineering
Georgia Tech

**Xiuyuan Cheng**
Department of Mathematics
Duke University

**Yao Xie**
School of Industrial and
Systems Engineering
Georgia Tech

## Abstract

Normalizing flow is a class of deep generative models for efficient sampling and likelihood estimation, which achieves attractive performance, particularly in high dimensions. The flow is often implemented using a sequence of invertible residual blocks. Existing works adopt special network architectures and regularization of flow trajectories. In this paper, we develop a neural ODE flow network called JKO-iFlow, inspired by the Jordan-Kinderleherer-Otto (JKO) scheme, which unfolds the discrete-time dynamic of the Wasserstein gradient flow. The proposed method stacks residual blocks one after another, allowing efficient block-wise training of the residual blocks, avoiding sampling SDE trajectories and score matching or variational learning, thus reducing the memory load and difficulty in end-to-end training. We also develop adaptive time reparameterization of the flow network with a progressive refinement of the induced trajectory in probability space to improve the model accuracy further. Experiments with synthetic and real data show that the proposed JKO-iFlow network achieves competitive performance compared with existing flow and diffusion models at a significantly reduced computational and memory cost.

## 1 Introduction

Generative models have wide applications in statistics and machine learning to infer data-generating distributions and to sample from the model distributions learned from the data. In addition to widely used deep generative models such as variational auto-encoders (VAE) Kingma and Welling [2014, 2019] and generative adversarial networks (GAN) Goodfellow et al. [2014], Gulrajani et al. [2017], normalizing flow models Kobyzev et al. [2020] have been popular and with a great potential. The flow model learns the data distribution via an invertible mapping $F$ between the data density $p_X$ in $\mathbb{R}^d$ and the target standard multivariate Gaussian density $p_Z$, $Z \sim \mathcal{N}(0, I_d)$. While flow models, once trained, can be utilized for efficient data sampling and explicit likelihood evaluation, training of such models is often difficult in practice. To alleviate such difficulties, many prior works [Dinh et al., 2015, 2017, Kingma and Dhariwal, 2018, Behrmann et al., 2019, Ruthotto et al., 2020, Onken et al., 2021], among others, have explored designs of training objectives, network architectures, and computational techniques.

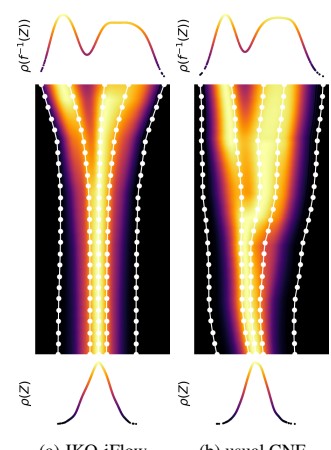

Figure 1: Comparison of JKO-iFlow (proposed) and standard CNF models. In contrast to most existing CNF models, JKO-iFlow learns the unique deterministic transport equation corresponding to the diffusion process by directly performing block-wise training of a neural ODE model.

Among various flow models, continuous normalizing flow (CNF) transports the data density to a target distribution through continuous dynamics, primarily neural ODE [Chen et al., 2018] models. CNF

37th Conference on Neural Information Processing Systems (NeurIPS 2023).

models have shown promising performance on generative tasks Grathwohl et al. [2019], Kobyzev et al. [2020]. However, a known computational challenge of CNF models is model regularization, primarily due to the non-uniqueness of the flow transport. Without additional regularization, the trained CNF model such as FFJORD Grathwohl et al. [2019] may follow a less regular trajectory in the probability space, see Figure 1(b), which may worsen the generative performance. While regularization of flow models has been developed using different techniques, including using spectral normalization Behrmann et al. [2019] and optimal transport Liutkus et al. [2019], Onken et al. [2021], Finlay et al. [2020], Xu et al. [2022], Huang et al. [2023], only using regularization may not resolve the non-uniqueness of the flow. Besides regularization, several practical difficulties remain when training CNF models, particularly the high computational cost. In many settings, CNF models consist of stacked blocks, and each can be complex. *End-to-end training* of such deep models often places a high demand on computational resources and memory consumption.

In this work, we propose JKO-iFlow, an invertible normalizing flow network that unfolds the Wasserstein gradient flow via a neural ODE model inspired by the JKO-scheme Jordan et al. [1998]. The JKO scheme, see (5), can be viewed as a proximal step to minimize the Kullback–Leibler (KL) divergence between the current density and the equilibrium density. It recovers the solution of the Fokker-Planck equation (FPE) in the limit of small step size. The proposed JKO-iFlow model thus can be viewed as trained to learn the *unique* transport map following the FPE which flows from the data distribution toward the normal equilibrium and gives a smooth trajectory of density evolution, see Figure 1(a). Unlike most CNF models, where all the residual blocks are trained end-to-end, each block in the JKO-iFlow network implements one step in the JKO scheme to learn the deterministic transport map by minimizing an objective of that block given the trained previous blocks. The block-wise training significantly reduces memory and computational load. Theoretically, with a small step size, the discrete-time transport map approximates the continuous solution of FPE, which leads to the invertibility of each trained residual block. This leaves the residual blocks to be completely general, such as graph neural network layers and convolutional neural network layers, depending on the structure of data considered in the problem. The theoretical need for small step sizes does not incur a restriction in practice, whereby one can use step size not exceeding a certain maximum value when adopting the neural ODE integration. We further introduce time reparameterization with progressive refinement in computing the flow network, where each block corresponds to a representative point along the density evolution trajectory in the space of probability measures. The algorithm adaptively chooses the number of blocks and step sizes.

The proposed approach is related to the diffusion models [Song and Ermon, 2019, Ho et al., 2020, Block et al., 2020, Song et al., 2021] yet differs fundamentally in that our approach is a type of flow models, which directly computes the data likelihood, and such likelihood estimation is essential for statistical inference. While the diffusion models can also indirectly obtain likelihood estimation, they are more designed as samplers. In terms of implementation, our approach trains a neural ODE model without SDE sampling (injection of noise) nor learning of score matching. It also differs from previous works on progressive training of ResNet generative models Johnson and Zhang [2019], Fan et al. [2022] in that the model trains an invertible flow mapping and avoids inner loops of variational learning. We refer to Section 1.1 for more discussions on related works. Empirically, JKO-iFlow yields competitive performance as other CNF models with significantly less computation. The model is also compatible with general equilibrium density $p_Z$ having a parametrized potential $V$, exemplified by the application to the conditional generation setting where $p_Z$ is replaced with Gaussian mixtures Xu et al. [2022]. In summary, the contributions of the work include

• We propose an invertible neural ODE model where each residual block corresponds to a JKO step, and the training objective can be computed from pushed data samples through the previous blocks. The residual block has a general form, and the invertibility is ensured due to the regularity and continuity of the approximate solution of the FPE.

• We develop a block-wise procedure to train the JKO-iFlow model, which can adaptively determine the number of blocks. We also propose to adaptively reparameterize the computed trajectory in the probability space with refinement, which improves the model accuracy and the overall computational efficiency.

• We show that JKO-iFlow greatly reduces memory and computational cost when achieving competitive or better generative performance and likelihood estimation compared to existing flow and diffusion models on simulated and real data.

## 1.1 Related works

For deep generative models, popular approaches include generative adversarial networks (GAN) [Goodfellow et al., 2014, Gulrajani et al., 2017, Isola et al., 2017] and variational auto-encoder (VAE)[Kingma and Welling, 2014, 2019]. Apart from known training difficulties (e.g., mode collapse [Salimans et al., 2016] and posterior collapse [Lucas et al., 2019]), these models do not provide likelihood or inference of data density. The normalizing flow framework [Kobyzev et al., 2020] has been extensively developed, including continuous flow [Grathwohl et al., 2019], Monge-Ampere flow [Zhang et al., 2018], discrete flow [Chen et al., 2019], and extension to non-Euclidean data [Liu et al., 2019, Mathieu and Nickel, 2020, Xu et al., 2022]. Efforts have been made to develop novel invertible mapping structures [Dinh et al., 2017, Papamakarios et al., 2017] and regularize the flow trajectories by transport cost  [Finlay et al., 2020, Onken et al., 2021, Ruthotto et al., 2020, Xu et al., 2022, Huang et al., 2023]. Despite such efforts, the model and computational challenges of normalizing flow models include regularization and the large model size when using a large number of residual blocks, which cannot be determined a priori, and the associated memory and computational load.

In parallel to continuous normalizing flows, which are neural ODE models, neural SDE models become an emerging tool for generative tasks. Diffusion process and Langevin dynamics in deep generative models have been studied in score-based generative models [Song and Ermon, 2019, Ho et al., 2020, Block et al., 2020, Song et al., 2021] under different settings. Specifically, these models estimate the score function (i.e., the gradient of the log probability density with respect to data) of data distribution via neural network parametrization, which may encounter challenges in learning and sampling high dimensional data and call for special techniques [Song and Ermon, 2019]. The recent work of Song et al. [2021] developed reverse-time SDE sampling for score-based generative models and adopted the connection to neural ODE to compute the likelihood; using the same idea of backward SDE, Zhang and Chen [2021] proposed joint training of forward and backward neural SDEs. Theoretically, latent diffusion Tzen and Raginsky [2019b,a] was used to analyze neural SDE models. The current work focuses on a neural ODE model where the deterministic vector field $\mathbf{f}(x, t)$ is to be learned from data following a JKO scheme of the FPE. Rather than neural SDE, our approach involves no sampling of SDE trajectories nor learning of the score function, and it learns an invertible residual network directly. In contrast, diffusion-based models derive the ODE model from the learned diffusion model to achieve explicit likelihood computation. For example, Song et al. [2021] derived neural ODE model from the learned score function of the diffused data marginal distributions for all $t$. We experimentally obtain competitive or improved performance against the diffusion model on simulated two-dimensional and high-dimensional tabular data.

JKO-inspired deep models have been studied in several recent works. [Bunne et al., 2022] reformulated the JKO step for minimizing an energy function over convex functions. JKO scheme has also been used to discretize Wasserstein gradient flow to learn a deep generative model in [Alvarez-Melis et al., 2022, Mokrov et al., 2021], which adopted input convex neural networks (ICNN) [Amos et al., 2017]. ICNN, as a special type of network architecture, may have limited expressiveness [Rout et al., 2022, Korotin et al., 2021]. In addition to using the gradient of ICNN, [Fan et al., 2022] proposed parametrizing the transport in a JKO step by a residual network but identified difficulty in calculating the push-forward distribution. The approach in [Fan et al., 2022] also relies on a variational formulation, which requires training an additional network similar to the discriminator in GAN using inner loops. The idea of progressive additive learning in training generative ResNet, namely training ResNet block-wisely by a variational loss, dates back to Johnson and Zhang [2019] under the GAN framework. Our method trains an invertible neural-ODE flow network that flows from data density to the normal one and backward, which enables the computation of model likelihood as in other neural-ODE approaches. The objective in each JKO step to minimize KL divergence can also be computed directly without any inner-loop training, see Section 4.

Compared to score-based neural SDE methods, our approach is closer to the more recent flow-based models related to diffusion models [Lipman et al., 2023, Albergo and Vanden-Eijnden, 2023, Boffi and Vanden-Eijnden, 2023]. These works proposed to learn the transport equation (a deterministic ODE) corresponding to the SDE process. Specifically, [Lipman et al., 2023, Albergo and Vanden-Eijnden,

2023] matched the velocity field from interpolated distributions between initial and terminal ones; [Boffi and Vanden-Eijnden, 2023] proposed to learn the score $\nabla \log \rho_t(x)$ (where $\rho_t$ solves the FPE and is unknown *a priori*) to solve high-dimensional FPE. Using Stein's identity (which is equivalent to the derivation in Section 3.2), the step-wise training objective in [Boffi and Vanden-Eijnden, 2023] optimizes to learn the score without the need to simulate the entire SDE. The idea of approximating the solution of FPE by a deterministic transport equation dates back to the 90s Degond and Mustieles [1990], Degond and Mas-Gallic [1989], and has been used in kernel-based solver of FPE in [Maoutsa et al., 2020] and studied via a self-consistency equation in [Shen et al., 2022]. While our approach learns an equivalent velocity field at the infinitesimal time step, see section 3.2, the formulation in JKO-iFlow is at a finite time-step motivated by the JKO scheme. We also mention that an independent concurrent work [Vidal et al., 2023] proposed a similar block-wise training algorithm using the JKO scheme under the framework of [Onken et al., 2021] and demonstrated benefits in avoiding tuning the penalty hyperparameter associated with the KL-divergence objective. Our model is applied to generative tasks of high-dimensional data, including image data, and we also develop additional techniques for computing the flow probability trajectory.

For the expressiveness of deep generative models, approximation properties of deep neural networks for representing probability distributions have been developed in several works. Lee et al. [2017] established approximation by composition of Barron functions [Barron, 1993]; Bailey and Telgarsky [2018] developed space-filling approach, which was generalized in Perekrestenko et al. [2020, 2021]; Lu and Lu [2020] constructed a deep ReLU network with guaranteed approximation under integral probability metrics, using techniques of empirical measures and optimal transport. These results show that deep neural networks can provably transport one source distribution to a target one with sufficient model capacity under certain regularity conditions of the pair of densities. Our JKO-iFlow model potentially leads to a constructive approximation analysis of the neural ODE flow model.

## 2 Preliminaries

*Normalizing flow.* A normalizing flow can be mathematically expressed via a density evolution equation of $\rho(x, t)$ such that $\rho(x, 0) = p_X$ and as $t$ increases $\rho(x, t)$ approaches $p_Z \sim \mathcal{N}(0, I_d)$ [Tabak and Vanden-Eijnden, 2010]. Given an initial distribution $\rho(x, 0)$, such a flow typically is not unique. We consider when the flow is induced by an ODE of $x(t)$ in $\mathbb{R}^d$

$$\dot{x}(t) = \mathbf{f}(x(t), t), \tag{1}$$

where $x(0) \sim p_X$. The marginal density of $x(t)$ is denoted as $p(x, t)$, and it evolves according to the continuity equation (Liouville equation) of (1) written as

$$\partial_t p + \nabla \cdot (p\mathbf{f}) = 0, \quad p(x, 0) = p_X(x). \tag{2}$$

*Ornstein–Uhlenbeck (OU) process.* Consider a Langevin dynamic denoted by the SDE $dX_t = -\nabla V(X_t)dt + \sqrt{2}dW_t$, where $V$ is the potential of the equilibrium density $p_Z$. We focus on the case of normal equilibrium, that is, $V(x) = |x|^2/2$ and then $p_Z \propto e^{-V}$. In this case, the process is known as the (multivariate) OU process. Suppose $X_0 \sim p_X$, and let the density of $X_t$ be $\rho(x, t)$ also denoted as $\rho_t(\cdot)$. The Fokker-Planck equation (FPE) describes the evolution of $\rho_t$ towards the equilibrium $p_Z$ as follows, where $V(x) := |x|^2/2$,

$$\partial_t \rho = \nabla \cdot (\rho \nabla V + \nabla \rho), \quad \rho(x, 0) = p_X(x). \tag{3}$$

Under generic conditions, $\rho_t$ converges to $p_Z$ exponentially fast. For Wasserstein-2 distance and the standard normal $p_Z$, classical argument gives that (take $C = 1$ in Eqn (6) of Bolley et al. [2012])

$$W_2(\rho_t, p_Z) \le e^{-t} W_2(\rho_0, p_Z), \quad t > 0. \tag{4}$$

*JKO scheme.* The seminal work Jordan et al. [1998] established a time discretization scheme of the solution to (3) by the gradient flow to minimize $\text{KL}(\rho||p_Z)$ under the Wasserstein-2 metric in probability space. Denote by $\mathcal{P}$ the space of all probability densities on $\mathbb{R}^d$ with a finite second moment. The JKO scheme computes a sequence of distributions $p_k$, $k = 0, 1, \cdots$, starting from $p_0 = \rho_0 \in \mathcal{P}$. With step size $h > 0$, the scheme at the $k$-th step is written as

$$p_{k+1} = \arg\min_{\rho \in \mathcal{P}} F[\rho] + \frac{1}{2h} W_2^2(p_k, \rho), \tag{5}$$

where $F[\rho] := \text{KL}(\rho||p_Z)$. It was proved in Jordan et al. [1998] that as $h \to 0$, the solution $p_k$ converges to the solution $\rho(\cdot, kh)$ of (3) for all $k$, and the convergence $\rho_{(h)}(\cdot, t) \to \rho(\cdot, t)$ is strongly in $L^1(\mathbb{R}^d, (0, T))$ for finite $T$ where $\rho_{(h)}$ is piece-wise constant interpolated from $p_k$.

# 3 JKO scheme by neural ODE

Given i.i.d. observed data samples $X_i \in \mathbb{R}^d$, $i = 1, \ldots, N$, drawn from some unknown density $p_X$, the goal is to train an invertible neural network to transport the density $p_X$ to an *a priori* specified density $p_Z$ in $\mathbb{R}^d$, where each data sample $X_i$ is mapped to a code $Z_i$. A prototypical choice of $p_Z$ is the standard multivariate Gaussian $\mathcal{N}(0, I_d)$. In this work, we leave the potential of $p_Z$ abstract and denote by $V$, that is, $p_Z \propto e^{-V}$ and $V(x) = |x|^2/2$ for normal $p_Z$. By a slight abuse of notation, we denote by $p_X$ and $p_Z$ both the distributions and the density functions of data $X$ and code $Z$ respectively.

## 3.1 Objective of JKO step

We are to specify $\mathbf{f}(x, t)$ in the ODE (1), to be parametrized and learned by a neural ODE, such that the induced density evolution of $p(x, t)$ converges to $p_Z$ as $t$ increases. We start by dividing the time horizon $[0, T]$ into finite subintervals with step size $h$, let $t_k = kh$ and $I_{k+1} := [t_k, t_{k+1})$. Define $p_k(x) := p(x, kh)$, namely the density of $x(t)$ at $t = kh$. The solution of (1) determined by the vector-field $\mathbf{f}(x, t)$ on $t \in I_{k+1}$ (assuming the ODE is well-posed [Sideris, 2013]) gives a one-to-one mapping $T_{k+1}$ on $\mathbb{R}^d$, s.t. $T_{k+1}(x(t_k)) = x(t_{k+1})$ and $T_{k+1}$ transports $p_k$ into $p_{k+1}$, i.e., $(T_k)_{\#} p_{k-1} = p_k$, where we denote by $T_{\#} p$ the push-forward of distribution $p$ by $T$, such that $(T_{\#} p)(A) = p(T^{-1}(A))$ for a measurable set $A$. In other words, the mapping $T_{k+1}$ is the solution map of the ODE from time $t_k$ to $t_{k+1}$.

Suppose we can find $\mathbf{f}(\cdot, t)$ on $I_{k+1}$ such that the corresponding $T_{k+1}$ solves the JKO scheme (5), then with small $h$, $p_k$ approximates the solution to the Fokker-Planck equation 3, which then flows towards $p_Z$. By the Monge formulation of the Wasserstein-2 distance between $p$ and $q$ as $W_2^2(p, q) = \min_{T: T_{\#} p = q} \mathbb{E}_{x \sim p} \|x - T(x)\|^2$, solving for the transported density $p_k$ by (5) is equivalent to solving for the transport $T_{k+1}$ by

$$T_{k+1} = \arg \min_{T: \mathbb{R}^d \to \mathbb{R}^d} F[T] + \frac{1}{2h} \mathbb{E}_{x \sim p_k} \|x - T(x)\|^2, \tag{6}$$

where $F[T] = \mathrm{KL}(T_{\#} p_k \| p_Z)$. The equivalence between (5) and (6) is proved in Lemma A.1.

Furthermore, the following proposition gives that, once $p_k$ is determined by $\mathbf{f}(x, t)$ for $t \leq t_k$, the value of $F[T]$ can be computed from $\mathbf{f}(x, t)$ on $t \in I_{k+1}$ only. The counterpart for convex function-based parametrization of $T_k$ was given in Theorem 1 of [Mokrov et al., 2021], where the computation using the change-of-variable differs as we adopt an invertible neural ODE approach here. The proof is left to Appendix A.

**Proposition 3.1.** *Given $p_k$, up to a constant $c$ independent from $\mathbf{f}(x, t)$ on $t \in I_{k+1}$,*

$$\mathrm{KL}(T_{\#} p_k \| p_Z) = \mathbb{E}_{x(t_k) \sim p_k} \left( V(x(t_{k+1})) - \int_{t_k}^{t_{k+1}} \nabla \cdot \mathbf{f}(x(s), s) ds \right) + c. \tag{7}$$

By Proposition 3.1, the minimization (6) is equivalent to

$$\min_{\{\mathbf{f}(x,t)\}_{t \in I_{k+1}}} \mathbb{E}_{x(t_k) \sim p_k} \left( V(x(t_{k+1})) - \int_{t_k}^{t_{k+1}} \nabla \cdot \mathbf{f}(x(s), s) ds + \frac{1}{2h} \|x(t_{k+1}) - x(t_k)\|^2 \right), \tag{8}$$

where $x(t_{k+1}) = x(t_k) + \int_{t_k}^{t_{k+1}} \mathbf{f}(x(s), s) ds$. Taking a neural ODE approach, we parametrize $\{\mathbf{f}(x, t)\}_{t \in I_{k+1}}$ as a residual block with parameter $\theta_{k+1}$, and then (8) is reduced to minimizing over $\theta_{k+1}$. This leads to a block-wise learning algorithm to be introduced in Section 4, where we further allow the step-size $h$ to vary for different $k$ as well.

## 3.2 Infinitesimal optimal $\mathbf{f}(x, t)$

In each JKO step of (8), let $p = p_k$ denote the current density, $q = p_Z$ be the target equilibrium density. In this subsection, we show that the optimal $\mathbf{f}$ in (8) with small $h$ reveals the difference between score functions between target and current densities. Thus, minimizing the objective (8) searches for a neural network parametrization of the score function $\nabla \log \rho_t$ implicitly, in contrast to diffusion-based models which learn the score function explicitly [Ho et al., 2020, Song et al.,

2021], e.g., via denoising score matching. At infinitesimal $h$, this is equivalent to solving the FPE by learning a deterministic transport equation as in [Boffi and Vanden-Eijnden, 2023, Shen et al., 2022].

Consider general equilibrium distribution $q$ with a differentiable potential $V$. To analyze the optimal pushforward mapping in the small $h$ limit, we shift the time interval $[kh, (k+1)h]$ to be $[0, h]$ to simplify the notation. Then (8) is reduced to

$$\min_{\{\mathbf{f}(x,t)\}_{t \in [0,h)}} \mathbb{E}_{x(0) \sim p} \left( V(x(h)) - \int_0^h \nabla \cdot \mathbf{f}(x(s), s) ds + \frac{1}{2h} \|x(h) - x(0)\|^2 \right), \quad (9)$$

where $x(h) = x(0) + \int_0^h \mathbf{f}(x(s), s) ds$. In the limit of $h \to 0+$, formally, $x(h) - x(0) = h\mathbf{f}(x(0), 0) + O(h^2)$, and suppose $V$ of $q$ is $C^2$, $V(x(h)) = V(x(0)) + h\nabla V(x(0)) \cdot \mathbf{f}(x(0), 0) + O(h^2)$. For any differentiable density $\rho$, the (Stein) score function is defined as $\mathbf{s}_\rho = \nabla \log \rho$, and we have $\nabla V = -\mathbf{s}_q$. Taking the formal expansion of orders of $h$, the objective in (9) is written as

$$\mathbb{E}_{x \sim p} \left( V(x) + h \left( -\mathbf{s}_q(x) \cdot \mathbf{f}(x, 0) - \nabla \cdot \mathbf{f}(x, 0) + \frac{1}{2} \|\mathbf{f}(x, 0)\|^2 \right) + O(h^2) \right). \quad (10)$$

Note that $\mathbb{E}_{x \sim p} V(x)$ is independent of $\mathbf{f}(x, t)$, and the $O(h)$ order term in (10) is over $\mathbf{f}(x, 0)$ only, thus the minimization of the leading term is equivalent to

$$\min_{\mathbf{f}(\cdot) = \mathbf{f}(\cdot, 0)} \mathbb{E}_{x \sim p} \left( -T_q \mathbf{f} + \frac{1}{2} \|\mathbf{f}\|^2 \right), \quad T_q \mathbf{f} := \mathbf{s}_q \cdot \mathbf{f} + \nabla \cdot \mathbf{f}, \quad (11)$$

where $T_q$ is known as the Stein operator [Stein, 1972]. The $T_q \mathbf{f}$ in (11) echoes that the derivative of KL divergence with respect to transport map gives Stein operator [Liu and Wang, 2016]. The Wasserstein-2 regularization gives an $L^2$ regularization in (11). Let $L^2(p)$ be the $L^2$ space on $(\mathbb{R}^d, p(x)dx)$, and for vector field $\mathbf{v}$ on $\mathbb{R}^d$, $\mathbf{v} \in L^2(p)$ if $\int |\mathbf{v}(x)|^2 p(x) dx < \infty$. One can verify that, when both $\mathbf{s}_p$ and $\mathbf{s}_q$ are in $L^2(p)$, the minimizer of (11) is

$$\mathbf{f}^*(\cdot, 0) = \mathbf{s}_q - \mathbf{s}_p.$$

This shows that the infinitesimal optimal $\mathbf{f}(x, t)$ equals the difference between the score functions of the equilibrium and the current density.

### 3.3 Invertibility of flow model and expressiveness

At time $t$ the current density of $x(t)$ is $\rho_t$, the analysis in Section 3.2 implies that the optimal vector field $\mathbf{f}(x, t)$ has the expression as

$$\mathbf{f}(x, t) = \mathbf{s}_q - \mathbf{s}_{\rho_t} = -\nabla V - \nabla \log \rho_t. \quad (12)$$

With this $\mathbf{f}(x, t)$, the Liouville equation (2) coincides with the FPE (3). This is consistent with the JKO scheme with a small $h$ recovering the solution to the FPE. Under proper regularity condition of $V$ and the initial density $\rho_0$, the r.h.s. of (12) is also regular over space and time. This leads to two consequences, in approximation and in learning: Approximation-wise, the regularity of $\mathbf{f}(x, t)$ allows to construct a $k$-th residual block in the flow network to approximate $\{\mathbf{f}(x, t)\}_{t \in I_k}$ when there is sufficient model capacity, by classical universal approximation theory of shallow networks [Barron, 1993, Yarotsky, 2017]. We further discuss the approximation analysis based on the proposed model in the last section.

For learning, when properly trained with sufficient data, the neural ODE vector field $\mathbf{f}(x, t; \theta_k)$ will learn to approximate (12). This can be viewed as inferring the score function of $\rho_t$, and also leads to the invertibility of the trained flow net in theory: Suppose the trained $\mathbf{f}(x, t; \theta_k)$ is close enough to (12); it will also have bounded Lipschitz constant. Then the residual block is invertible as long as the step size $h$ is sufficiently small, e.g. less than $1/L$ where $L$ is the Lipschitz bound of $\mathbf{f}(x, t; \theta_k)$. In practice, we typically use smaller $h$ than needed merely by invertibility (allowed by the model budget) so that the flow network can more closely track the FPE of the diffusion process. The invertibility of the proposed model is numerically verified in experiments (see Table A.1).

## 4 Training of JKO-iFlow net

The proposed JKO-iFlow model allows progressive learning of the residual blocks in the neural-ODE model in a block-wise manner (Section 4.1). We also introduce two techniques to improve the training of the trajectories in probability space (Section 4.2), illustrated in a vector space in Appendix B.3.

### 4.1 Block-wise training

Note that the training of $(k+1)$-th block in (8) can be conducted once the previous $k$ blocks are trained. Specifically, with finite training data $\{X_i = x_i(0)\}_{i=1}^n$, the expectation $\mathbb{E}_{x(t)\sim p_k}$ in (8) is replaced by the sample average over $\{x_i(kh)\}_{i=1}^n$ which can be computed from the previous $k$ blocks. Note that for each given $x(t) = x(t_k)$, both $x(t_{k+1})$ and the integral of $\nabla \cdot \mathbf{f}$ in (8) can be computed by a numerical neural ODE integrator. Following previous works, we use the Hutchinson trace estimator [Hutchinson, 1989, Grathwohl et al., 2019] to compute the quantity $\nabla \cdot \mathbf{f}$ in high dimensions, and we also propose a finite-difference approach to reduce the computational cost (details in Appendix B.2). Applying the numerical integrator in computing (8), we denote the resulting $k$-th residual block abstractly as $f_{\theta_k}$ with trainable parameters $\theta_k$.

This leads to a block-wise training of the normalizing flow network, as summarized in Algorithm 1. The sequence of time stamps $t_k$ is given by specifying the time steps $h_k := t_{k+1} - t_k$, which we allow to differ across $k$. The choice of the sequence $h_k$ is initialized by a geometric sequence starting from $h_0$ with maximum stepsize $h_{\max}$, see Appendix B.1. In the special case where the multiplying factor is one, the sequence of $h_k$ gives a constant step size. The adaptive choice of $h_k$ with refinement (by adding more blocks) will be introduced in Section 4.2. Regarding the termination criterion $\text{Ter}(\mathrm{k})$ in line 2 of Algorithm 1, we monitor the ratio $\mathbb{E}_{x\sim p_{k-1}}\|x - T_k(x)\|^2/\mathbb{E}_{x\sim p_{k-1}}\|T_k(x)\|^2$ and terminate when it is below some threshold $\epsilon$, set as 0.01 in all experiments. In practice, when training the $k$-th block, both the averages of $\|x - T_k(x)\|^2$ and $\|T_k(x)\|^2$ are computed by empirically averaging over the training samples (in the last epoch) at no additional computational cost. Lastly, line 5 of training a "free block" (i.e., the block without the $W_2$ regularization) is to flow the push-forward density $p_L$ closer to the target density $p_Z$, where the former is obtained through the first $L$ blocks.

The block-wise training significantly reduces the memory and computational load since only one block is trained when optimizing (8) regardless of flow depth. Therefore, one can use larger training batches and potentially more expensive numerical integrators within a certain memory budget for higher accuracy. We also empirically observe that training each block using standard back-propagation (rather than the adjoint method in neural ODE) gives a comparable result at a lower cost. To ensure the invertibility of the trained JKO-iFlow network, we further break the time interval $[t_{k-1}, t_k)$ into 3 or 5 subintervals to compute the neural ODE integration, e.g., by Runge-Kutta-4. We empirically verify small inversion errors on test samples.

---

**Algorithm 1** Block-wise JKO-iFlow training

---

**Require:** Time stamps $\{t_k\}$, training data, termination criterion $\text{Ter}$ and tolerance level $\epsilon$, maximal number of blocks $L_{\max}$.
1: Initialize $k = 1$.
2: **while** $\text{Ter}(\mathrm{k}) > \epsilon$ and $k \leq L_{\max}$ **do**
3:     Optimize $f_{\theta_k}$ upon minimizing (8) with mini-batch sample approximation, given $\{f_{\theta_i}\}_{i=1}^{k-1}$. Set $k \leftarrow k + 1$.
4: **end while**
5: $L \leftarrow k$. Optional: Optimize $f_{\theta_{L+1}}$ without $W_2$ regularization.

---

### 4.2 Computation of trajectories in probability space

We adopt two additional computational techniques to facilitate learning of the trajectories in the probability space, represented by the sequence of densities $p_k$, $k = 1, \ldots, L$, associated with the $L$ residual blocks of the proposed normalizing flow network. The two techniques are illustrated in Figure 2. Further details and illustrations of the approach can be found in Appendix B.

• *Trajectory reparameterization.* We empirically observe fast decay of the movements $W_2^2(T_\# p_k, p_k)$ when $h_k$ is set to be constant, that is, initial blocks transport the densities much further than the later ones. This is consistent with the exponential convergence of the Fokker-Planck flow, see (4), but unwanted in the algorithm because in order to train the current block, the flow model needs to transport data through all previous blocks, and yet the later blocks trained using constant step size barely contribute to the density transport. Hence, instead of having constant $h_k$, we *reparameterize* the values of $t_k$ through an adaptive procedure based on the $W_2$ distance at each block. The procedure uses an adaptive approach to encourage the $W_2$ movement in each block to be more even across the $L$ blocks, where the retraining of the trajectory can be potentially warm-started by the previous trajectory in iteration.

• *Progressive refinement.* The performance of CNF models is typically improved with a larger number of residual blocks, corresponding to a smaller step size. The smallness of stepsize $h$ also ensures the invertibility of the flow model, in theory and also in practice. However, directly training the model with a small non-adaptive stepsize $h$ may result in long computation time and convergence to the normal density $q_Z$ only after a large number of blocks, where the choice of $h_k$ is not as efficient as after adaptive reparameterization. We introduce a refinement approach

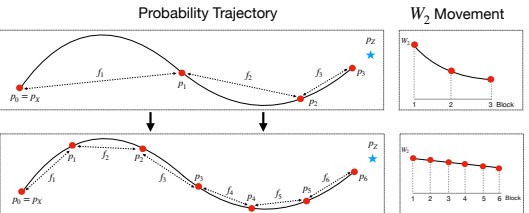

Figure 2: Diagram illustrating trajectory reparameterization and refinement. Top: the original trajectory under three blocks via Algorithm 1. Bottom: the trajectory under six blocks after reparameterization and refinement, which renders the $W_2$ movements more even.

to increase the number of blocks progressively, where each time interval $[t_{k-1}, t_k)$ splits into two halves, and the number of blocks doubles after the adaptive reparameterization of the trajectory converges at the coarse level. The new trajectory at the refined level is again trained with adaptive reparameterization, where the residual blocks can be warm-started from the coarse-level model to accelerate the convergence. The trajectory refinement allows going to a smaller step size $h_k$, which benefits the accuracy of the JKO-iFlow model including the numerical accuracy in integrating each neural ODE block.

## 5 Experiment

In this section, we examine the proposed JKO-iFlow model on simulated and real datasets, including both unconditional and conditional generation tasks. Codes are available at `https://github.com/hamrel-cxu/JKO-iFlow`.

### 5.1 Baselines and metrics

We compare five alternatives, including four CNF models and one diffusion model. The first two CNF models are FFJORD [Grathwohl et al., 2019] and OT-Flow [Onken et al., 2021], which are continuous-time flow (neural-ODE models). The next two CNF models are IResNet [Behrmann et al., 2019] and IGNN [Xu et al., 2022], which are discrete in time (ResNet models). The diffusion model baseline is the score-based neural SDE [Song et al., 2021], which we call "ScoreSDE." Details about the experimental setup, including dataset construction and neural network architecture and training can be found in Appendix C.2.

The accuracy of trained generative models is evaluated by two quantitative metrics, the negative log-likelihood (NLL) metric, and the kernel *maximum mean discrepancy* (MMD) [Gretton et al., 2012a] metric, including MMD-1, MMD-m, and MMD-c for constant, median distance, and custom bandwidth respectively. The test threshold $\tau$ is computed by bootstrap, where an MMD metric less than $\tau$ indicates that the generated distribution is evaluated by the MMD test to be the same as the true data distribution (achieving $\alpha = 0.05$ test level). See details in Appendix C.1. The computational cost is measured by the number of mini-batch stochastic gradient descent steps and the training

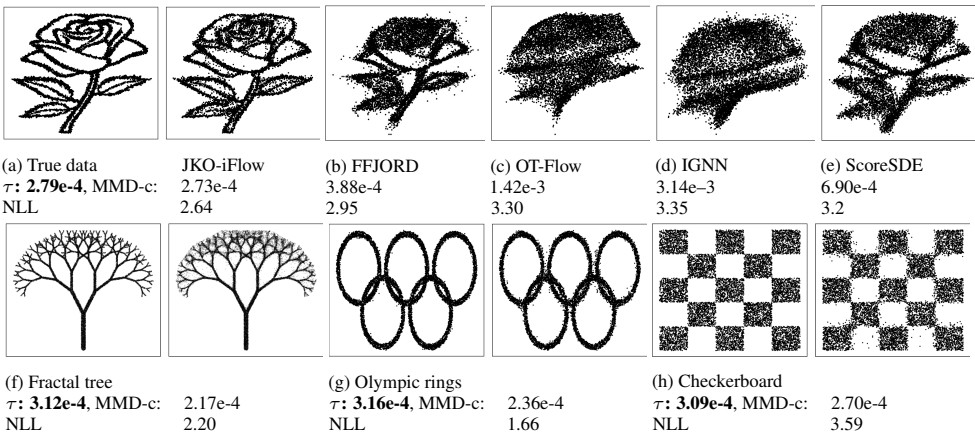

Figure 3: Results on two-dimensional simulated datasets by JKO-iFlow and competitors.

Table 1: Results on tabular datasets. All competitors are trained in a fixed-budget setup using 10 times more mini-batches (their performances using the same number of mini-batches are worse and not comparable to JKO-iFlow). See the complete table in Table A.2. The results using more computation are given in Table A.3.

| Data Set | Model | # Param | Test MMD-m | Test MMD-1 | NLL | Data Set | Model | # Param | Test MMD-m | Test MMD-1 | NLL |
|---|---|---|---|---|---|---|---|---|---|---|---|
| | | | $\tau$: **1.73e-4** | $\tau$: **2.90e-4** | | | | | $\tau$: **2.46e-4** | $\tau$: **3.75e-4** | |
| **POWER** $d=6$ | JKO-iFlow | 76K | 9.86e-5 | 2.40e-4 | -0.12 | **MINIBOONE** $d=43$ | JKO-iFlow | 112K | 9.66e-4 | 3.79e-4 | 12.55 |
| | OT-Flow | 76K | 7.58e-4 | 5.35e-4 | 0.32 | | OT-Flow | 112K | 6.58e-4 | 3.79e-4 | 11.44 |
| | FFJORD | 76K | 9.89e-4 | 1.16e-3 | 0.63 | | FFJORD | 112K | 3.51e-4 | 4.12e-4 | 23.77 |
| | IGNN | 304K | 1.93e-3 | 1.59e-3 | 0.95 | | IGNN | 448K | 1.21e-2 | 4.01e-4 | 26.45 |
| | IResNet | 304K | 3.92e-3 | 2.43e-2 | 3.37 | | IResNet | 448K | 2.13e-3 | 4.16e-4 | 22.36 |
| | ScoreSDE | 76K | 9.12e-4 | 6.08e-3 | 3.41 | | ScoreSDE | 112K | 5.86e-1 | 4.33e-4 | 27.38 |
| | | | $\tau$: **1.85e-4** | $\tau$: **2.73e-4** | | | | | $\tau$: **1.38e-4** | $\tau$: **1.01e-4** | |
| **GAS** $d=8$ | JKO-iFlow | 76K | 1.52e-4 | 5.00e-4 | -7.65 | **BSDS300** $d=63$ | JKO-iFlow | 396K | 2.24e-4 | 1.91e-4 | -153.82 |
| | OT-Flow | 76K | 1.99e-4 | 5.16e-4 | -6.04 | | OT-Flow | 396K | 5.43e-1 | 6.49e-1 | -104.62 |
| | FFJORD | 76K | 1.87e-3 | 3.28e-3 | -2.65 | | FFJORD | 396K | 5.60e-1 | 6.76e-1 | -37.80 |
| | IGNN | 304K | 6.74e-3 | 1.43e-2 | -1.65 | | IGNN | 990K | 5.64e-1 | 6.86e-1 | -37.68 |
| | IResNet | 304K | 3.20e-3 | 2.73e-2 | -1.17 | | IResNet | 990K | 5.50e-1 | 5.50e-1 | -33.11 |
| | ScoreSDE | 76K | 1.05e-3 | 8.36e-4 | -3.69 | | ScoreSDE | 396K | 5.61e-1 | 6.60e-1 | -7.55 |

time. The performance is also reported under a fixed-budget setting to compare across models. The numerical inversion error of the trained flow models are at the 1e-5 level or below, see Table A.1.

## 5.2 Two-dimensional toy data

The results on four toy datasets are shown in Figure 3, where the metrics of NLL and MMD-c are reported. Plots (a)-(e) compare JKO-iFlow with the alternative models on the dataset Rose, where both NLL and MMD-c metrics show the better performance of JKO-iFlow. Visually, the generated samples $\hat{X}$ by JKO-iFlow also more resemble the true data distribution of $X$ than the competitors in (b)-(e). Plots (f)-(h) show the results by JKO-iFlow on the examples of Fractal tree, Olympic rings, and Checkerboard, where JKO-iFlow gives a satisfactory generative performance. The results of JKO-iFlow in Figure 3 are obtained after applying the trajectory improvement techniques in Section 4.2. The comparison results without using the techniques are provided in Figures A.2 and A.3, which show the benefit of the two techniques in learning the model.

## 5.3 High-dimensional tabular data

Table 1 assesses the performance of JKO-iFlow and baseline methods on four high-dimensional tabular datasets under a fixed-budget setting: we keep the model size of continuous flow models (OT-Flow and FFJORD) and the diffusion model (ScoreSDE) the same, and we increase model sizes for discrete flow models (IGNN and IResNet) which can be computed faster therein and this also improves their performances. Except for MINIBOONE dataset, JKO-iFlow yields smaller or similar quantitative metrics compared to all alternatives. The visual comparisons of generated data distributions are shown in Figure A.4.

After additional training to apply the trajectory reparametrization, the performance of JKO-iFlow is furtherly improved, see Table A.3 in comparison with additional baselines of OT-Flow and FFJORD from the original references. The effects of trajectory reparameterization on the MINIBOONE dataset are shown in Figure A.5, where the $W_2$ movements across the blocks become more even as the iteration proceeds, see (a). The generative performance, as shown in (b), visually improves after the reparametrization, which is consistent with the lower NLL values in Table A.3 (NLL improves from 12.55 to 10.55). These results show that JKO-iFlow performs comparably to the best baseline, and often better under small computational and memory budgets.

## 5.4 Image generation

We apply the JKO-iFlow model to image generation tasks on the MNIST, CIFAR-10, and Imagenet-32 datasets. In all examples, we train JKO-iFlow in the latent space of a pre-trained variational auto-encoder (VAE) adopted from [Esser et al., 2021]. Training generative models in latent space have been shown to obtain state-of-the-art image generation performance, e.g., in StableDiffusion [Rombach et al., 2022]. Here we train a flow model instead of a score-based diffusion model in the latent space, and more details can be found in Appendix C.2.

The generated images are shown in Figure 4. The quantitative evaluation is by the Fréchet inception distance (FID) score [Heusel et al., 2017], which is included in the figure captions. Comparatively, we are aware of some performance gap between our model and the state-of-the-art performance of score-based diffusion models like DDPM [Ho et al., 2020] and ScoreSDE [Song et al., 2021], however, the results here are obtained with less computation. Specifically, on a single A100 GPU, our experiments took 24 hours on CIFAR10 and 30 hours on Imagenet-32. Meanwhile, the image generation by JKO-iFlow model here obtains visually more appealing images and achieves lower

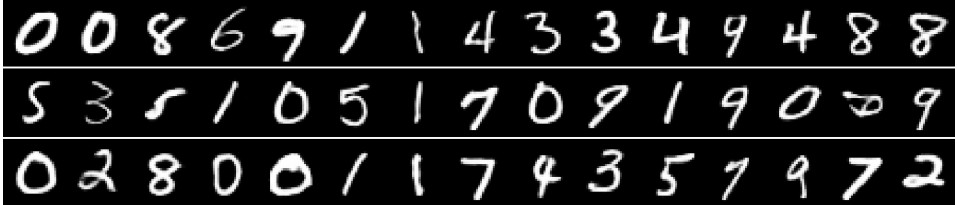

(a) Generated MNIST digits. FID: 7.95.

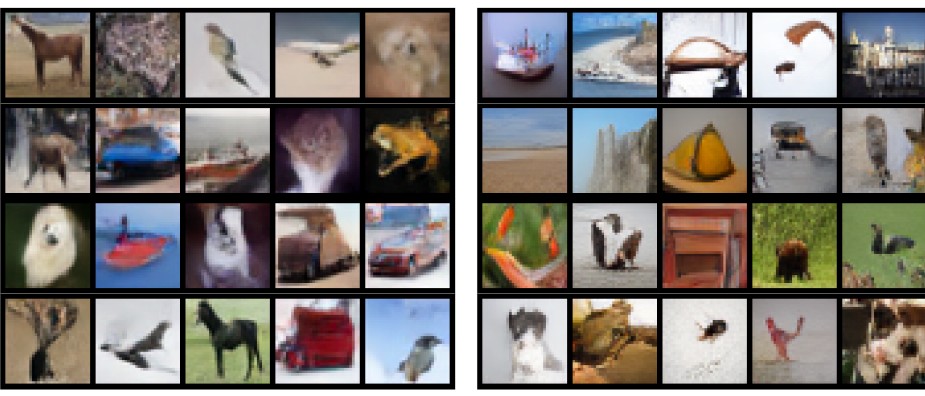

(b) Generated CIFAR10 images. FID: 29.10.    (c) Generated Imagenet-32 images. FID: 20.10.

Figure 4: Generated samples of MNIST, CIFAR10, and Imagenet-32 by JKO-iFlow model in latent space. We select 2 images per class for CIFAR10 and 1 image per class for Imagenet-32. The FIDs are shown in subcaptions. Uncurated samples are shown in Figure A.6.

FIDs compared to most CNF baselines [Grathwohl et al., 2019, Behrmann et al., 2019, Chen et al., 2019, Finlay et al., 2020].

### 5.5 Conditional generation

The problem aims to generate input samples $X$ given a label $Y$ from the conditional distribution $X|Y$ to be learned from data. We follow the approach in IGNN [Xu et al., 2022]. In this setting, the JKO-iFlow network pushes from the distribution $X|Y = k$ to the class-specific component in the Gaussian mixture of $H|Y = k$, see Figure A.7b and Appendix C.2.4 for more details. We apply JKO-iFlow to the Solar ramping dataset and compare it with the original IGNN model, and both models use graph neural network layers in the residual blocks. The results are shown in Figure A.8, where both the NLL and MMD-m metrics indicate the superior performance of JKO-iFlow and is consistent with the visual comparison.

## 6 Discussion

The work can be extended in several directions. The application to larger-scale image datasets and larger graphs will enlarge the scope of usage. To overcome the computational challenge faced by neural ODE models for high dimensional input, e.g., images of higher resolution, one would need to improve the training efficiency of the backpropagation in neural ODE in addition to the dimension reduction techniques by VAE as been explored here. Another possibility is to combine the JKO-iFlow scheme with other backbone flow models that are more suitable for the specific tasks. Meanwhile, it would be interesting to extend the method to other problems for which CNF models have proven to be effective. Examples include multi-dimensional probabilistic regression [Chen et al., 2018], a plug-in to deep architectures such as StyleFlow [Abdal et al., 2021], and the application to Mean-field Games [Huang et al., 2023].

Theoretically, the expressiveness of the flow model to generate a regular data distribution can be analyzed based on Section 3.3. To sketch a road map, a block-wise approximation guarantee of $\mathbf{f}(x, t)$ as in (12) can lead to approximation of the Fokker-Planck flow (3), which pushes forward the density to be $\varepsilon$-close to normality in $T = \log(1/\varepsilon)$ time, see (4). Reversing the time of the ODE then leads to an approximation of the initial density $\rho_0 = p_X$ by flowing backward in time from $T$ to zero. Further analysis under technical assumptions is left to future work. During the time that this paper was being published, the convergence analysis of the proposed model was studied in Cheng et al. [2023].

## Acknowledgement

The authors thank Jianfeng Lu, Yulong Lu, and Yiping Lu for helpful discussions. The work is partially supported by NSF DMS-2134037. C.X. and Y.X. are partially supported by an NSF CAREER CCF-1650913, and NSF DMS-2134037, CMMI-2015787, CMMI-2112533, DMS-1938106, and DMS-1830210 and the Coca-Cola Foundation. XC is also partially supported by NSF DMS-2237842 and Simons Foundation.

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

# A Proofs

## A.1 Proofs in Section

**Lemma A.1.** *Suppose $p$ and $q$ are two densities on $\mathbb{R}^d$ in $\mathcal{P}$, the following two problems*

$$\min_{\rho \in \mathcal{P}} L_\rho[\rho] = \text{KL}(\rho||q) + \frac{1}{2h} W_2^2(p, \rho), \tag{13}$$

$$\min_{T: \mathbb{R}^d \to \mathbb{R}^d} L_T[T] = \text{KL}(T_\# p||q) + \frac{1}{2h} \mathbb{E}_{x \sim p}||x - T(x)||^2, \tag{14}$$

*have the same minimum and*

*(a) If $T^* : \mathbb{R}^d \to \mathbb{R}^d$ is a minimizer of (14), then $\rho^* = (T^*)_\# p$ is a minimizer of (13).*

*(b) If $\rho^*$ is a minimizer of (13), then the optimal transport from $p$ to $\rho^*$ minimizes (14).*

*Proof of Lemma A.1.* Let the minimum of (14) be $L_T^*$, and that of (13) be $L_\rho^*$.

Proof of (a): Suppose $L_T$ achieves minimum at $T^*$, then $T^*$ is the optimal transport from $p$ to $\rho^* = (T^*)_\# p$ because otherwise $L_T$ can be further improved. By definition of $L_\rho$, we have $L_T^* = L_T[T^*] = L_\rho[\rho^*] \geq L_\rho^*$. We claim that $L_T^* = L_\rho^*$. Otherwise, there is another $\rho'$ such that $L_\rho[\rho'] < L_T^*$. Let $T'$ be the optimal transport from $p$ to $\rho'$, and then $L_T[T'] = L_\rho[\rho'] < L_T^*$, contradicting with that $L_T^*$ is the minimum of $L_T$. This also shows that $L_\rho[\rho^*] = L_T^* = L_\rho^*$, that is, $\rho^*$ is a minimizer of $L_\rho$.

Proof of (b): Suppose $L_\rho$ achieves minimum at $\rho^*$. Let $T^*$ be the OT from $p$ to $\rho^*$, then $\mathbb{E}_{x \sim p}|x - T^*(x)|^2 = W_2(p, \rho^*)^2$, and then $L_T[T^*] = L_\rho[\rho^*] = L_\rho^*$ which equals $L_T^*$ as proved in (a). This shows that $T^*$ is a minimizer of $L_T$. $\qquad\square$

*Proof of Proposition 3.1,* Given $p_k$ being the density of $x(t)$ at $t = kh$, recall that $T$ is the solution map from $x(t)$ to $x(t + h)$. We denote $\rho_t := p_k$, and $\rho_{t+h} := T_\# p_k$. By definition,

$$\text{KL}(T_\# p_k || p_Z) = \mathbb{E}_{x \sim \rho_{t+h}}(\log \rho_{t+h}(x) - \log p_Z(x)). \tag{15}$$

Because $p_Z \propto e^{-V}$, $V(x) = |x|^2/2$, we have $\log p_Z(x) = -V(x) + c_1$ for some constant $c_1$. Thus

$$\mathbb{E}_{x \sim \rho_{t+h}} \log p_Z(x) = \mathbb{E}_{x(t) \sim \rho_t} \log p_Z(x(t + h)) = c_1 - \mathbb{E}_{x(t) \sim \rho_t} V(x(t + h)). \tag{16}$$

To compute the first term in (15), note that

$$\mathbb{E}_{x \sim \rho_{t+h}} \log \rho_{t+h}(x) = \mathbb{E}_{x(t) \sim \rho_t} \log \rho_{t+h}(x(t + h)), \tag{17}$$

and by the expression (called "instantaneous change-of-variable formula" in normalizing flow literature [Chen et al., 2018], which we derive directly below)

$$\frac{d}{dt} \log \rho(x(t), t) = -\nabla \cdot \mathbf{f}(x(t), t), \tag{18}$$

we have that for each value of $x(t)$,

$$\log \rho_{t+h}(x(t + h)) = \log \rho(x(t + h), t + h) = \log \rho(x(t), t) - \int_t^{t+h} \nabla \cdot \mathbf{f}(x(s), s) ds.$$

Inserting back to (17), we have

$$\mathbb{E}_{x \sim \rho_{t+h}} \log \rho_{t+h}(x) = \mathbb{E}_{x(t) \sim \rho_t} \log \rho_t(x(t)) - \mathbb{E}_{x(t) \sim \rho_t} \int_t^{t+h} \nabla \cdot \mathbf{f}(x(s), s) ds.$$

The first term is determined by $\rho_t = p_k$, and thus is a constant $c_2$ independent from $\mathbf{f}(x, t)$ on $t \in [kh, (k + 1)h]$. Together with (16), we have shown that

$$\text{r.h.s. of (15)} = c_2 - \mathbb{E}_{x(t) \sim \rho_t} \int_t^{t+h} \nabla \cdot \mathbf{f}(x(s), s) ds - c_1 + \mathbb{E}_{x(t) \sim \rho_t} V(x(t + h)),$$

which proves (7).

Derivation of (18): by chain rule,

$$
\begin{aligned}
\frac{d}{dt} \log \rho(x(t), t) &= \frac{\nabla \rho(x(t), t) \cdot \dot{x}(t) + \partial_t \rho(x(t), t)}{\rho(x(t), t)} \\
&= \left. \frac{\nabla \rho \cdot \mathbf{f} - \nabla \cdot (\rho \mathbf{f})}{\rho} \right|_{(x(t), t)} \quad \text{(by (1) and (2))} \\
&= -\nabla \cdot \mathbf{f}(x(t), t).
\end{aligned}
$$

$\square$

# B  Technical details of Section 4

To train the proposed JKO-iFlow model by Algorithm 1, one needs to specify a sequence of $t_k$, where the $k$-th JKO block integrates from $t_{k-1}$ to $t_k$, starting from $t_0 = 0$. Denote by $h_k := t_{k+1} - t_k$ the step-size for $k$-th step.

## B.1  Initial choice of $t_k$ in Algorithm 1

We first initialize the choice by the following scheme

$$
h_k = \min\{\rho^k h_0, h_{\max}\}, \quad k = 0, 1, \cdots \tag{19}
$$

where the base stepsize $h_0$, the multiplying factor $\rho \geq 1$, and the maximum stepsize $h_{\max} \geq h_0$ are three hyper-parameters to be specified by the user. When $\rho = 1$, the sequence of $h_k$ in (19) corresponds to constant stepsize $h_k \equiv h_0$.

Applying Algorithm 1 with the initial choice of $t_k$ gives $L$ trained residual blocks of the neural ODE model, where $L$ is to be determined by the stopping criterion in Section 4.1. The model is further improved by the two additional techniques in Section 4.2, which will adaptively specify the stepsize $h_k$ and we explain in detail below.

## B.2  Hutchinson trace via finite difference

We propose a finite-difference estimator of $\nabla \cdot \mathbf{f}(x, t) = \mathrm{Tr}(J_{\mathbf{f}}(x))$, where $J_{\mathbf{f}}(x))$ is the Jacobian of $\mathbf{f}$ at $x$. This is based on the Hutchinson trace estimator [Hutchinson, 1989], which states that $\mathrm{Tr}(J_{\mathbf{f}}(x)) = \mathbb{E}_{p(\epsilon)} \left[ \epsilon^T J_{\mathbf{f}}(x) \epsilon \right]$. Here, $p(\epsilon)$ is a distribution in $\mathbb{R}^d$ satisfying $\mathbb{E}[\epsilon] = 0$ and $\mathrm{Cov}(\epsilon) = I$ (e.g., a standard Gaussian). Existing CNF works estimate the expectation by sampling $\epsilon \sim p(\epsilon)$ and computing the vector-Jacobian product $J_{\mathbf{f}}(x)\epsilon$ using reverse-mode automatic differentiation. The product $J_{\mathbf{f}}(x)\epsilon$ can be computed for approximately the same cost as evaluating $\mathbf{f}$ [Grathwohl et al., 2019].

Now, given a fixed $\epsilon$, we have $J_{\mathbf{f}}(x)\epsilon = \lim_{\sigma \to 0} \frac{\mathbf{f}(x + \sigma\epsilon) - \mathbf{f}(x)}{\sigma}$, which is the directional derivative of $\mathbf{f}$ along the direction $\epsilon$. Thus, given a fixed small $\sigma > 0$, we propose the following finite-difference estimator of $\nabla \cdot \mathbf{f}(x, t)$:

$$
\nabla \cdot \mathbf{f}(x, t) \approx \mathbb{E}_{p(\epsilon)} \left[ \epsilon^T \frac{\mathbf{f}(x + \sigma\epsilon) - \mathbf{f}(x)}{\sigma} \right]. \tag{20}
$$

The finite-difference approximation of $J_{\mathbf{f}}(x)\epsilon$ requires only one additional function evaluation of $\mathbf{f}$ at $x + \sigma\epsilon$ and avoids the computation of automatic differentiation. We empirically found that training with (20) on high-dimensional examples (e.g., $d > 100$ as in image examples) can be approximately 1.5~2 times faster than existing CNF approaches relying on reverse-mode automatic differentiation. In all experiments, we let $\sigma = \sigma_0 / \sqrt{d}$ with $\sigma_0 = 0.02$.

## B.3  Trajectory improvement illustrated in vector space

Section 4.2 introduces two techniques to improve the computation of a gradient descent trajectory represented by discrete points. For illustrative purposes, we introduce the two techniques when the gradient system is in a vector space equipped with Euclidean metric. The JKO-iFlow training applies the same technique in the gradient system equipped with the Wasserstein-2 metric.

Let the space be $\mathbb{R}^d$, and $F : \mathbb{R}^d \to \mathbb{R}$ be a differential landscape. Given a sequence of positive stepsize $h_k$, one can compute a sequence of representative points $x_k$ which discretizes the gradient descent trajectory of minimizing $F$ (towards a local minimum $x^*$). Specifically, we have

$$x_{k+1} = \arg\min_x F(x) + \frac{1}{2h_k}\|x - x_k\|_2^2, \tag{21}$$

starting from some fixed $x_0$.

### B.3.1 Trajectory reparameterization

Starting from an initial sequence of step-size $\{h_k\}_{k=1}^L$ as in Section B.1, the reparameterization technique implements an iterative scheme to update the sequence of $h_k$ adaptively. We call the $j$-th iteration 'Iter-$j$', and denote the sequence by $\{h_k^{(j)}\}_{k=1}^L$. The corresponding solution points $x_k$ by solving (21) with $h_k^{(j)}$ are denoted as $x_k^{(j)}$. In addition to the constant $h_{\max}$, we need another algorithmic parameter $\eta \in (0,1)$, and we set $\eta = 0.3$ for the vector-space example in computation.

We always have $x_0^{(j)} = x_0$ for all 'Iter-$j$'. In the $j$-th iteration of the parametrization, we compute the following

1. Compute the *arclength* as

$$S_k^{(j)} := \|x_{k-1}^{(j)} - x_k^{(j)}\|_2, \quad k = 1, \cdots, L. \tag{22}$$

2. Compute the average arclength

$$\bar{S} := \sum_{k=1}^L S_k^{(j)}/L.$$

3. Update the stepsize for $k = 1, \ldots, L$ as

$$h_k^{(j+1)} := \min\{h_k^{(j)} + \eta(\bar{S}h_k^{(j)}/S_k^{(j)} - h_k^{(j)}), h_{\max}\}$$

4. Solve (21) using $h_k^{(j+1)}$ to obtain $x_k^{(j+1)}$.

We terminate the iteration when the arclength $\{S_k^{(j)}\}$ are approximately equal. An illustration is given as Iter-12 in the upper panel of Figure A.1. After the reparameterization iterations, we solve an additional $x_{L+1}$ by minimizing $F(x)$ starting from $x_L$. This corresponds to optimizing the "free-block" in Algorithm 1.

### B.3.2 Progressive refinement

The scheme can be computed using a positive integer refinement factor. For simplicity, we use factor 2 throughout this work.

Given a sequence of $\{h_k\}_{k=1}^L$ and the representative points $\{x_k\}_{k=1}^L$, the refinement scheme computes a refined trajectory having $k = 1, \cdots, 2L$:

1. Compute the refined stepsizes as

$$\tilde{h}_{2k-1} = \tilde{h}_{2k} = h_k/2, \quad k = 1, \ldots, L$$

2. Compute the representative points $\tilde{x}_k$ by solving (21) using $\tilde{h}_k$, possibly warm-starting by $\tilde{x}_{2k} = x_k$ and $\tilde{x}_{2k-1} = (x_k + x_{k-1})/2$.

We then apply the trajectory reparameterization iterations as in Section B.3.1 to the refined trajectory till convergence, and the free-block ending point is also recomputed. An illustration is given as r-Iter-10 in the upper panel of Figure A.1.

### B.4 Trajectory improvement in probability space

We apply the two techniques to solve the JKO step (8), which is the counterpart of (21) as a gradient descent scheme with proximal steps. The representative points $x_k$ are replaced with transported distributions $p_k$ which form a sequence of points in $\mathcal{P}$, and the optimization is over the parametrization of $p_k$ induced by the neural network flow mapping consisting of the first $k$ residual blocks.

It remains to define the arclength in (22) to implement the two techniques. Because we equip $\mathcal{P}$ with the Wasserstein-2 metric, we compute (omitting the iteration index $j$ in the notation)

$$S_k = W_2(p_{k-1}, p_k) = (\mathbb{E}_{x \sim p_{k-1}} \|x - T_k(x)\|^2)^{1/2}, \tag{23}$$

where the transport mapping $T_k(x) = x + \int_{t_{k-1}}^{t_k} f_{\theta_k}(x(s), s) ds$ and can be computed from the $k$-th block. In practice, the expectation $\mathbb{E}_{x \sim p_{k-1}}$ in (23) is computed by a finite-sample average on the training set.

At last, the optimal warm-start of $p_k$ in the refinement is implemented by inheriting the parameters $\theta_k$ of the trained blocks.

## C Experimental details

### C.1 Quantitative evaluation metrics

Besides visual comparison, we adopt two quantitative metrics to evaluate the performance of generative models, the negative log-likelihood (NLL) metric Grathwohl et al. [2019] and the maximum mean discrepancy (MMD) [Gretton et al., 2012a] metric.

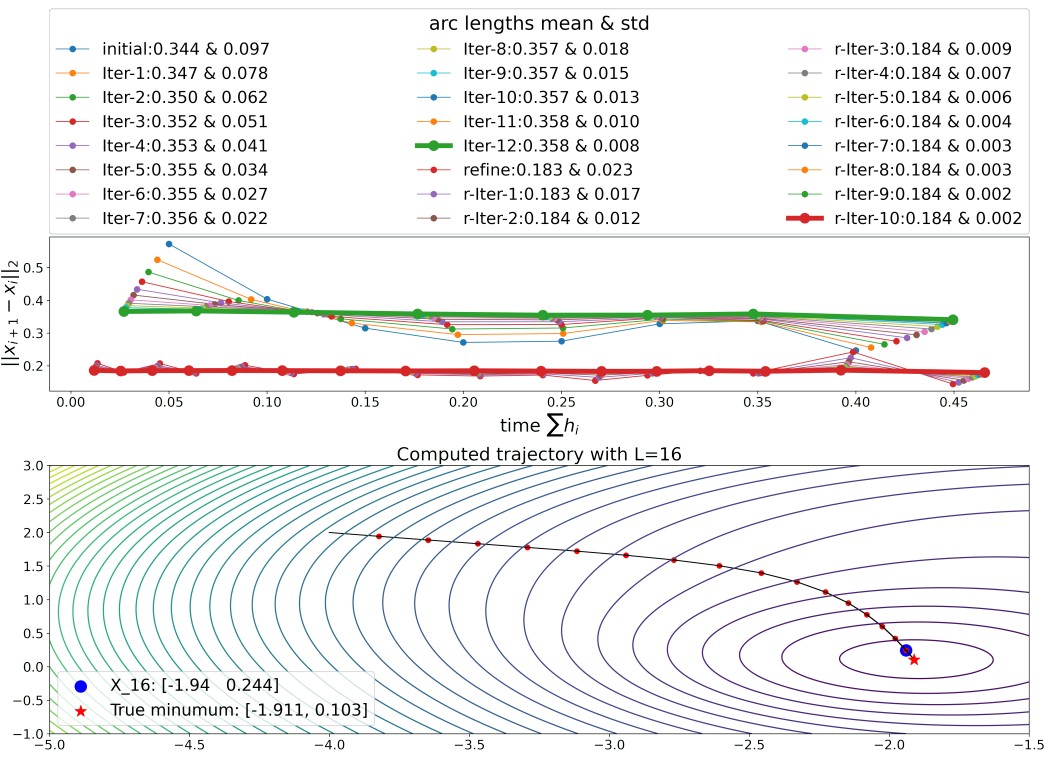

Figure A.1: The upper panel shows the arc lengths and the mean and standard deviation of arc lengths over 12 reparameterization iterations, one refinement, and an additional 10 reparameterization iterations. The lower panel visualizes the trajectory consisting of 17 solution points at "r-Iter-10", where the free point as the 17-th point computes the approximate minimum by $x_{17} = x_{L+1} = [-1.911, 0.105]$.

### C.1.1 NLL metric

Our computation of the NLL metric follows the standard procedure for neural ODE normalizing flow models Chen et al. [2018], Grathwohl et al. [2019], where the evolution of density $\rho(x(t))$ can be computed via integrating $\nabla \cdot \mathbf{f}(x(t), t)$ over time due to the instantaneous change-of-variable formula (18).

Specifically, our model flows from $t_0 = 0$ to $t_{L+1} = T$, where $x(0) \sim p_0$ the data distribution, and we train the flow model to make $x(T)$ follow a normal density $q$. The model density at data sample $x$ is expressed as $\rho(x, 0) = (T_\theta^{-1})_\# q(x)$ where $T_\theta^{-1}$ is the inverse model flow mapping from $x(T)$ to $x(0)$. Thus the model log-likelihood can be expressed as

$$\log \rho(x(0), 0) = \log q(x(T), T) + \int_0^T \nabla \cdot \mathbf{f}(x(s), s) ds.$$

In practice, our trained flow model has $L$ blocks where each block $f_{\theta_k}$ represents $\mathbf{f}(x, t)$ on $[t_{k-1}, t_k]$ and is parametrized by $\theta_k$. The log-likelihood at test sample $x$ is then computed by

$$\text{LL}(x) = -\frac{1}{2}(\|x(t_{L+1})\|_2^2 + d \log(2\pi)) + \sum_{k=1}^{L+1} \int_{t_{k-1}}^{t_k} \nabla \cdot f_{\theta_k}(x(s), s) ds, \tag{24}$$

where

$$x(t_k) = x(t_{k-1}) + \int_{t_{k-1}}^{t_k} f_{\theta_k}(x(s), s) ds$$

starting from $x(0) = x$. Both the integration of $f_{\theta_k}$ and $\nabla \cdot f_{\theta_k}$ are computed using the numerical scheme of neural ODE. We report NLL in the natural unit of information (i.e., $\log$ with base $e$, known as "nats") in all our experiments.

### C.1.2 MMD metrics

Note that the normalizing flow models use NLL (on training set) as the training objective. In contrast, the MMD metric is an impartial evaluation metric as it is not used to train JKO-iFlow or any competing methods. Given two set of data samples $\boldsymbol{X} := \{x_i\}_{i=1}^N$ and $\tilde{\boldsymbol{X}} := \{\tilde{x}_j\}_{j=1}^M$ and a kernel function $k(x, \tilde{x})$, the (squared) kernel MMD [Gretton et al., 2012a] is defined as

$$\text{MMD}(\boldsymbol{X}, \tilde{\boldsymbol{X}}) := \frac{1}{N^2} \sum_{i=1}^N \sum_{j=1}^N k(x_i, x_j) + \frac{1}{M^2} \sum_{i=1}^M \sum_{j=1}^M k(\tilde{x}_i, \tilde{x}_j) - \frac{2}{NM} \sum_{i=1}^N \sum_{j=1}^M k(x_i, \tilde{x}_j), \tag{25}$$

When a generative model is trained, we generate $M$ i.i.d. data samples by the model to construct the set $\tilde{\boldsymbol{X}}$, and we form the set $\boldsymbol{X}$ using $N$ true data samples (from the test set). MMD metrics with other choices of kernels are possible Gretton et al. [2012b], Sutherland et al. [2017], Schrab et al. [2023]. In all experiments here, we use the Gaussian kernel $k(x, \tilde{x}) = \exp\{-\|x - \tilde{x}\|^2/2h^2\}$ to stay consistent with reported baselines from [Onken et al., 2021], where $h > 0$ is the bandwidth parameter. We use three ways of setting the bandwidth parameter $h$:

- Constant bandwidth: $h = h_c = 1$. The resulting MMD is denoted as 'MMD-1'.
- Median bandwidth [Gretton et al., 2012a]: let $h = h_m$ be the median of $\|x_i - x_j\|$ for all distinct $i$ and $j$. The median distance is computed from the dataset $X$. The resulting MMD is denoted as 'MMD-m'.
- Custom bandwidth: on certain datasets when we can use prior knowledge to decide on the bandwidth, we will custom the choice of $h$ (typically smaller than the median distance, due to that theoretically smaller bandwidth may lead to a more powerful MMD test to distinguish the difference in the underlying distributions) while ensuring that we use large enough $M$ and $N$ to compute the MMD metric. We call the metric 'MMD-c'.

On all datasets, we use at most $N = 10K$ test samples as $\mathbf{X}$, and for each trained model, we generate $M = 10K$ test samples to form the dataset $\tilde{\mathbf{X}}$ to compute the MMD value defined as in (25). Note that the MMD metric as a measure of distance between two distributions is significant when above a test threshold $\tau$, which is defined as the upper $(1 - \alpha)$-quantile of the distribution of the MMD

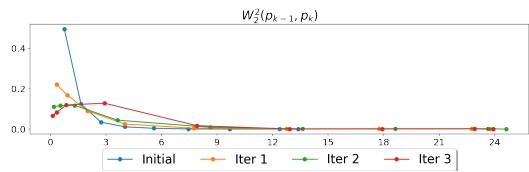

(a) Per-block $W_2^2$ over reparameterization iterations.

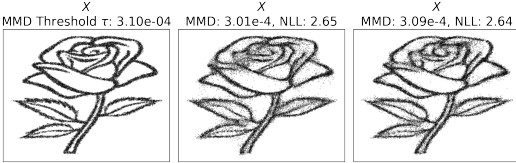

(b) Results at initial training (middle) and Iter 3 (right). MMD and NLL values are shown in the title.

Figure A.2: Same plots as in Figure A.5 for Rose data. We observe improved generative quality on the leaves of the rose after reparameterization iterations

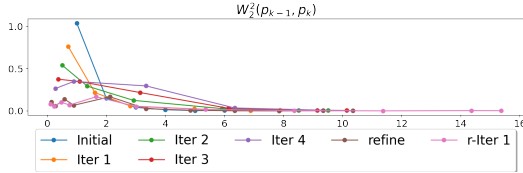

(a) Per-block $W_2^2$ over reparameterization iterations and refinement ('r-Iter 1' means one reparameterization iteration after refinement).

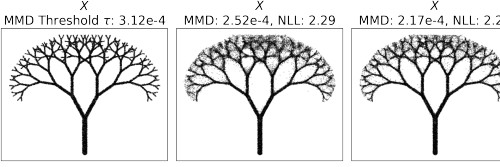

(b) Results at Iter 4 (middle) and r-Iter 1 (right). MMD and NLL values are shown in the title.

Figure A.3: Same plots as in Figure A.5 for Fractal tree data. We observe improved generative quality on the edges of the tree after refinement.

statistic under the null hypothesis (i.e., when dataset $\mathbf{X}$ and $\tilde{\mathbf{X}}$ observe the same distribution). The scalar $\alpha$ is the controlled Type-I error (known as the test level), which is set to be 0.05. To obtain the test threshold $\tau$ for the MMD, we adopt the bootstrap procedure Arcones and Gine [1992], Gretton et al. [2012a] and compute $\tau$ as the empirical $(1-\alpha)$-quantile of the simulated null distribution of the MMD from the pool of samples formed by the union of $\mathbf{X}$ and $\tilde{\mathbf{X}}$, where the set $\tilde{\mathbf{X}}$ is generated by the trained JKO-iFlow model. We use 1000 times of bootstrap in all experiments. In our usage of evaluating generative models, the threshold $\tau$ can be viewed as a baseline of the MMD metric, that is, when the computed MMD values are above $\tau$, then the smaller the MMD value the better the generative performance; when the computed MMD value is below $\tau$, it means that with respect to the current MMD metric, the trained model generates a data distribution that is as good as the true distribution.

## C.2  Detail setup and additional results

All experiments are conducted using PyTorch [Paszke et al., 2019] and PyTorch Geometric [Fey and Lenssen, 2019]. The optimizer is Adam [Kingma and Ba, 2015] with learning rates to be specified in each example. We use the neural-ODE integrator [Grathwohl et al., 2019, adjoint method] on all examples.

### C.2.1  Two-dimensional toy data

*Training and test data:* We generate 8K test samples for each dataset, and for the training split,

- For Rose (Figure 3 and A.2) and Checkerboard and Olympic rings (Figure 3), we re-sample 10K training samples every epoch, for a total of 100 epochs per block. The batch size is 500.
- For Fractal tree (Figure 3 and A.3), we use a fixed training data of 200K samples, for a total of 50 epochs per block. The batch size is 2000.

*Choice of $h_k$:* The initial schedule is by setting $h_0 = 0.75, \rho = 1.2$. We set $h_{\max} = 5$ for rose, checkerboard, and Olympic rings, and $h_{\max} = 3$ for fractal tree. For reparametrization and refinement, we use $\eta = 0.5$ in the reparameterization iterations for all examples.

Table A.1: Inversion error $\mathbb{E}_{x \sim p_X} \| T_\theta^{-1}(T_\theta(x)) - x \|_2^2$ of JKO-iFlow computed via sample average on the test split of the data set, where $T_\theta$ denotes the transport mapping over all the blocks of the trained flow network.

| Rose | Fractal tree | Olympic rings | Checkerboard | POWER | GAS | MINIBOONE | BSD300 | MNIST |
|------|--------------|---------------|--------------|-------|-----|-----------|--------|-------|
| 3.30e-6 | 3.58e-5 | 2.24e-6 | 3.07e-5 | 1.48e-5 | 1.58e-6 | 1.09e-6 | 1.53e-5 | 1.87e-5 |

Table A.2: Complete results on real tabular datasets to augment Table 1 under the fixed-budget setting. $L$ denotes to the number of residual blocks used in each dataset; for comparison, the free block in JKO-iFlow is not used. For fair comparison across models, the number of batches indicates how many batches pass through all residual blocks.

| Data Set | Model | # Param | Training | | | | Testing | | |
|---|---|---|---|---|---|---|---|---|---|
| | | | Time (h) | # Batches | Time/Batches (s) | Batch size | MMD-m | MMD-1 | NLL |
| **POWER** $d=6$ | | | | | | | $\tau$: **1.73e-4** | $\tau$: **2.90e-4** | |
| | JKO-iFlow | 76K, L=4 | 0.07 | 0.76K | 3.51e-1 | 10000 | 9.86e-5 | 2.40e-4 | -0.12 |
| | OT-Flow | 76K | 0.36 | 7.58K | 1.71e-1 | 10000 | 7.58e-4 | 5.35e-4 | 0.32 |
| | FFJORD | 76K, L=4 | 0.67 | 7.58K | 3.18e-1 | 10000 | 9.89e-4 | 1.16e-3 | 0.63 |
| | IGNN | 304K, L=16 | 0.29 | 7.58K | 1.38e-1 | 10000 | 1.93e-3 | 1.59e-3 | 0.95 |
| | IResNet | 304K, L=16 | 0.41 | 7.58K | 1.95e-1 | 10000 | 3.92e-3 | 2.43e-2 | 3.37 |
| | ScoreSDE | 76K | 0.06 | 7.58K | 2.85e-2 | 10000 | 9.12e-4 | 6.08e-3 | 3.41 |
| | ScoreSDE | 76K | 0.60 | 75.80K | 2.85e-2 | 10000 | 7.12e-4 | 5.04e-3 | 3.33 |
| | JKO-iFlow | 57K, L=3 | 0.05 | 0.76K | 2.63e-1 | 10000 | 3.86e-4 | 7.20e-4 | -0.06 |
| **GAS** $d=8$ | | | | | | | $\tau$: **1.85e-4** | $\tau$: **2.73e-4** | |
| | JKO-iFlow | 76K, L=4 | 0.07 | 0.76K | 3.32e-1 | 5000 | 1.52e-4 | 5.00e-4 | -7.65 |
| | OT-Flow | 76K | 0.23 | 7.60K | 1.09e-1 | 5000 | 1.99e-4 | 5.16e-4 | -6.04 |
| | FFJORD | 76K, L=4 | 0.65 | 7.60K | 3.08e-1 | 5000 | 1.87e-3 | 3.28e-3 | -2.65 |
| | IGNN | 304K, L=16 | 0.34 | 7.60K | 1.61e-1 | 5000 | 6.74e-3 | 1.43e-2 | -1.65 |
| | IResNet | 304K, L=16 | 0.46 | 7.60K | 2.18e-1 | 5000 | 3.20e-3 | 2.73e-2 | -1.17 |
| | ScoreSDE | 76K | 0.03 | 7.60K | 1.42e-2 | 5000 | 1.05e-3 | 8.36e-4 | -3.69 |
| | ScoreSDE | 76K | 0.30 | 76.00K | 1.42e-2 | 5000 | 2.23e-4 | 3.38e-4 | -5.58 |
| | JKO-iFlow | 95K, L=5 | 0.09 | 0.76K | 4.15e-1 | 5000 | 1.51e-4 | 3.77e-4 | -7.80 |
| **MINIBOONE** $d=43$ | | | | | | | $\tau$: **2.46e-4** | $\tau$: **3.75e-4** | |
| | JKO-iFlow | 112K, L=4 | 0.03 | 0.34K | 3.61e-1 | 2000 | 9.66e-4 | 3.79e-4 | 12.55 |
| | OT-Flow | 112K | 0.21 | 3.39K | 2.23e-1 | 2000 | 6.58e-4 | 3.79e-4 | 11.44 |
| | FFJORD | 112K, L=4 | 0.28 | 3.39K | 2.97e-1 | 2000 | 3.51e-3 | 4.12e-4 | 23.77 |
| | IGNN | 448K, L=16 | 0.63 | 3.39K | 6.69e-1 | 2000 | 1.21e-2 | 4.01e-4 | 26.45 |
| | IResNet | 448K, L=16 | 0.71 | 3.39K | 7.54e-1 | 2000 | 2.13e-3 | 4.16e-4 | 22.36 |
| | ScoreSDE | 112K | 0.01 | 3.39K | 6.37e-3 | 2000 | 5.86e-1 | 4.33e-4 | 27.38 |
| | ScoreSDE | 112K | 0.10 | 33.90K | 6.37e-3 | 2000 | 4.17e-3 | 3.87e-4 | 20.70 |
| **BSDS300** $d=63$ | | | | | | | $\tau$: **1.38e-4** | $\tau$: **1.01e-4** | |
| | JKO-iFlow | 396K, L=4 | 0.05 | 1.03K | 1.85e-1 | 1000 | 2.24e-4 | 1.91e-4 | -153.82 |
| | OT-Flow | 396K | 0.62 | 10.29K | 2.17e-1 | 1000 | 5.43e-1 | 6.49e-1 | -104.62 |
| | FFJORD | 396K, L=4 | 0.54 | 10.29K | 1.89e-1 | 1000 | 5.60e-1 | 6.76e-1 | -37.80 |
| | IGNN | 990K, L=10 | 1.71 | 10.29K | 5.98e-1 | 1000 | 5.64e-1 | 6.86e-1 | -37.68 |
| | IResNet | 990K, L=10 | 2.05 | 10.29K | 7.17e-1 | 1000 | 5.50e-1 | 5.50e-1 | -33.11 |
| | ScoreSDE | 396K | 0.01 | 10.29K | 3.50e-3 | 1000 | 5.61e-1 | 6.60e-1 | -7.55 |
| | ScoreSDE | 396K | 0.10 | 102.90K | 3.50e-3 | 1000 | 5.61e-1 | 6.62e-1 | -7.31 |
| | JKO-iFlow | 396K, L=4 | 0.08 | 1.03K | 2.76e-1 | 5000 | 1.41e-4 | 8.83e-5 | -156.68 |

- For Rose, 3 reparameterization moving iteration, no refinement.
- For Checkerboard and Olympic rings, 4 reparameterization moving iteration, no refinement.
- For Fractal tree, 4 reparameterization moving iteration, one refinement, and an additional reparameterization moving iteration.

*MMD metric:* We use custom kernel bandwidth $h = 0.1h_m$ where $h_m$ is the median distance of dataset $\mathbf{X}$ with $N = 8K$ (from true data distribution). From trained generative model, we generate $M = 10K$ test samples as $\tilde{\mathbf{X}}$.

*Network and activation:* Fully-connected residual blocks with two hidden layers. We use the softplus activation ($\beta = 20$) with 128 hidden dimensions. Before refinement, we train 9 residual blocks for rose, checkerboard, and Olympic rings, and train 6 residual blocks for fractal trees. The learning rate is 5e-3.

*Additional results:* Figure A.2 and A.3 illustrates the benefits of reparameterization and refinement. Specifically, the details of the generated rose around portions of leaves and of the generated tree around the smallest edges are clearer using these techniques. In terms of quantitative metrics, the NLL is also smaller, even though the MMD metric indicates there is no statistical difference between the ground truth and generated samples.

### C.2.2 Tabular datasets

*Training and test data:* The four high-dimensional real datasets (POWER, GAS, MINIBOONE, BSDS300) come from the University of California Irvine (UCI) machine learning data repository, and we follow the pre-processing procedures of [Papamakarios et al., 2017]. Regarding data sizes,

- POWER: 1.85M training sample and 205K test sample.
- GAS: 1M training sample, 100K test sample.

Table A.3: The OT-Flow and FFJORD baselines marked with * are from the original papers [Onken et al., 2021] and [Grathwohl et al., 2019]. The results of JKO-iFlow are obtained after applying the reparameterization technique.

| Data Set | Model | # Param | Training # Batches | Training Batch size | Testing NLL |
|---|---|---|---|---|---|
| **POWER** $d = 6$ | JKO-iFlow | 95K, L=5 | 6.08K | 10000 | -0.40 |
|  | OT-Flow* | 18K | 22K | 10000 | -0.30 |
|  | FFJORD* | 43K, L=5 | - | 10000 | -0.46 |
| **GAS** $d = 8$ | JKO-iFlow | 114K, L=6 | 6.08K | 5000 | -9.43 |
|  | OT-Flow* | 127K | 52K | 2000 | -9.20 |
|  | FFJORD* | 279K, L=5 | - | 1000 | -8.59 |
| **MINIBOONE** $d = 43$ | JKO-iFlow | 112K, L=4 | 2.72K | 2000 | 10.55 |
|  | OT-Flow* | 78K | 7K | 2000 | 10.55 |
|  | FFJORD* | 821K, L=1 | - | 1000 | 10.43 |
| **BSDS300** $d = 63$ | JKO-iFlow | 495K, L=5 | 2.06K | 5000 | -157.75 |
|  | OT-Flow* | 297K | 37K | 300 | -154.20 |
|  | FFJORD* | 6.7M, L=2 | - | 10000 | -157.40 |

- MINIBOONE: 32K training sample, 3.6K test sample.

- BSDS300: 1.05M training sample, 250K test sample.

*Choice of $h_k$:* The initial schedule is by setting $h_0 = 1, \rho = 1, h_{\max} = 3$. For reparametrization and refinement, we use $\eta = 0.5$ in the reparameterization iterations for all datasets.

*MMD metric:* When the test data from the MINIBOONE dataset is less than 10K in size, we use all the test sets as $\mathbf{X}$ and $N$ is the size of test data. When the test data from all three other datasets has more than 10K samples, we randomly select $N = 10K$ samples from the entire test data, and the same random test subset is used to evaluate all models. We use the median distance kernel bandwidth and generate $M = 10K$ test samples from each model to form $\tilde{\mathbf{X}}$.

*Network and activation:* We use the softplus activation for all networks. Regarding the design of residual blocks, we use fully-connected residual blocks with two hidden layers. On POWER, GAS, and MINIBOONE, we use 128 hidden nodes, and on BSDS300, we use 256 hidden nodes. The number of residual blocks for four tabular datasets is described in Tables A.2 and A.3. Regarding learning rate, it is 1e-3 on POWER, 2e-3 on GAS, 5e-3 on MINIBOONE, and 1.75e-3 on BSDS300.

*Additional results:* For experiments under the fixed-budget setting, Table A.2 shows the complete results of JKO-iFlow against competitors on tabular datasets. In the extra lines in the table, we show the result of JKO-iFlow with L=3 for POWER and L=5 for GAS, as these numbers of $L$ are determined by the termination criterion in Algorithm 1. (The main lines show the results with $L = 4$ so that our model has the same capacity as the alternatives.) BSDS300 has an extra line for JKO-iFlow trained with batch sizes 5000, which improves over the result with batch size 1000. For ScoreSDE, we use the implementation in [Huang et al., 2021], which, during training, maximizes the evidence lower bound (ELBO) as an equivalent objective to the score-matching loss. Although ScoreSDE is the fastest, its performance, even under 100 times more mini-batch stochastic gradient descent steps than JKO-iFlow, is still worse than JKO-iFlow in terms of both MMD-m and NLL.

To visualize the generative performance, scatter plots of the generated samples by JKO-iFlow and competitors on these tabular datasets are shown in Figure A.4 after projected to 2 dimensions (determined by principal components computed from true data test samples). The plots show a closer match between those from JKO-iFlow with the ground truth in distribution. For experiments allowing more expensive models and longer training time, Table A.3 shows the results of JKO-iFlow after additional training of trajectory reparametrization in comparison with other baselines cited from the original papers. We run 7 reparameterization iterations on POWER, GAS, and MINIBOONE and 1 reparameterization iteration on BSDS300. A free block is used on POWER, GAS, and BSDS300.

Table A.4: NLL per noise scheduler and $\bar{\beta}_{\max}$ combination of ScoreSDE on MINIBOONE. The three settings "linear, constant, quadratic" follow the DDPM suggestion [Ho et al., 2020]. The mean and standard deviation are computed over three replicas of the trained model.

| Noise scheduler \ $\beta_{\max}$ | 1 | 5 | 10 | 15 | 20 |
|---|---|---|---|---|---|
| Linear | 24.51 (0.24) | 18.73 (0.64) | 20.28 (0.38) | 26.76 (1.77) | 26.83 (1.23) |
| Constant | 25.47 (0.25) | 30.37 (0.84) | 37.73 (0.98) | 40.47 (1.27) | 45.32 (0.40) |
| Quadratic | 27.19 (0.14) | 17.45 (0.17) | 18.48 (0.38) | 18.90 (0.48) | 21.35 (0.60) |

Table A.5: Testing NLL per noise scheduler and $\bar{\beta}_{\max}$ combination of ScoreSDE on MINIBOONE, using a larger network with longer training. The table format is identical to that of Table A.4.

| Noise scheduler $\setminus \beta_{\max}$ | 5 | 10 | 15 |
|---|---|---|---|
| Linear | 11.33 (0.44) | 10.84 (1.17) | 10.47 (0.13) |
| Quadratic | 12.88 (0.40) | 11.11 (0.24) | 11.05 (0.49) |

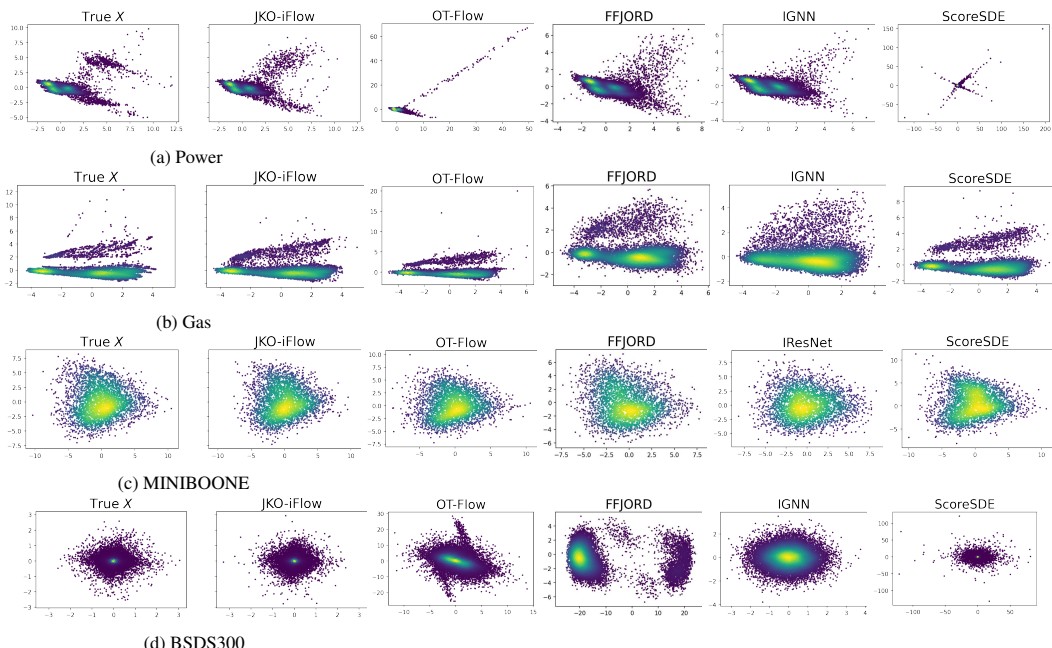

Figure A.4: Generative quality on tabular datasets via PCA projection of generated samples. The generative quality in general aligns with the quantitative metrics in Table 1 and A.2.

To ensure a fair comparison against diffusion models, we perform additional experiments using different noise schedulers $\beta(t)$ and $\bar{\beta}_{\max}$ in ScoreSDE on MINIBOONE, following the noise scheduler suggestions in DDPM [Ho et al., 2020]. Note that DDPM can be viewed as a discrete-time version of the variance-preserving ScoreSDE model. As shown in Table A.4, the performance of ScoreSDE is indeed sensitive to the noise schedule. However, the best NLL of ScoreSDE 17.45 from the table is still noticeably higher than the NLL of 12.55 obtained by JKO-iFlow on MINIBOONE in Table A.2, and JKO-iFlow is trained using ten times less number of batches. To improve the performance of ScoreSDE on this example, we use ScoreSDE without restrictions on modeling and computation. Specifically, we consider a larger network following the setup in [Albergo and Vanden-Eijnden, 2023], where the network has 4 hidden layers with 512 hidden nodes per layer and consists of 831K parameters in total. We then train this larger model for 100K batches using ScoreSDE with various noise schedulers, choosing noise schedulers that performed best based on results in Table A.4. From the results reported in Table A.5, we see that the testing NLL by ScoreSDE can be as low as 10.47, which is among the state-of-the-art values reported in Table A.3. Nevertheless, we want to highlight that the proposed JKO-iFlow can obtain competitive results (i.e., 10.55 in Table A.3) with much smaller models: each JKO-iFlow block has only 2 hidden layers with 128 hidden nodes per layer, and we trained 4 blocks that contain 112K parameters in total. We also trained JKO-iFlow for only 2.72K batches, rather than the 100K batches we used for ScoreSDE.

### C.2.3 Image generation with pre-trained variational auto-encoder

We perform image generation on MNIST [Deng, 2012], CIFAR10 [Krizhevsky and Hinton, 2009], and Imagenet-32 [Deng et al., 2009] datasets. We do so in the latent space of a pre-trained VAE, where we discuss the details below.

*VAE as data pre-processing:* We train deep VAEs in an adversarial manner following [Esser et al., 2021], and use pre-trained VAEs to pre-process the input images $X$. Specifically, the encoder $\mathcal{E} : X \rightarrow (\mu(X), \Sigma(X))$ of the VAE maps a RGB-image $X$ to parameters of a multivariate Gaussian distribution $\mathcal{N}(\mu(X), \Sigma(X))$ in a lower dimension $\tilde{d}$. Then, given any random latent code $\tilde{X} \sim \mathcal{N}(\mu(X), \Sigma(X))$, the decoder $\mathcal{D} : \tilde{X} \rightarrow \hat{X}$ of the VAE is trained so that $X \approx \hat{X}$ for the reconstructed image $\hat{X} = \mathcal{D}(\tilde{X})$. On MNIST, we let the latent-space dimension $\tilde{d} = 20$, and on CIFAR10 and ImageNet32, we let the latent-space dimension $\tilde{d} = 192$. There are 5.6M parameters in the VAE encoder and 8.25M parameters in the VAE decoder. Given a trained VAE, we then train the JKO-iFlow model $T_\theta$ to transport invertibly between the distribution of random latent codes $\tilde{X}$ defined over all training images $X$ and the standard multivariate Gaussian distribution in $\mathbb{R}^{\tilde{d}}$.

*Training and test data:* Training data: 60K images in MNIST, 50K images in CIFAR10, and 1.28M images in Imagenet-32. Test data: 10K images in MNIST and CIFAR10, and 50K images in Imagenet-32.

*Choice of $h_k$:* on MNIST, we specify $\{h_0 = 1, \rho = 1, h_{\max} = 1\}$. on CIFAR10, we specify $\{h_0 = 1, \rho = 1, h_{\max} = 1\}$. On Imagenet-32, we specify $\{h_0 = 1, \rho = 1.1, h_{\max} = \infty\}$. We train $L = 6$ blocks on MNIST, and we train $L = 8$ blocks on CIFAR10 and Imagenet-32. On CIFAR10 and Imagenet-32, we also scale $h_k = T \cdot h_k / \sum_j h_j$ so $\sum_k h_k = T$. We let $T = 0.8$ on CIFAR10 and $T = 0.4$ on Imagenet-32.

*Network architecture:* We use the softplus activation with $\beta = 20$ for all hidden layers. On MNIST, we use fully connected residual blocks with three hidden layers at 256 hidden nodes. On CIFAR10 and Imagenet-32, we parametrize each $\mathbf{f}_{\theta_b}$ as a concatenation of convolution layers and transposed convolution layers. Specifically:

- CIFAR10: convolution layers have channels 3-64-128-256-256 with kernel size 3 and strides 1-1-2-1. Transposed convolution layers have channels 256-256-128-64-3 with kernel size 3-4-3-3 and strides 1-2-1-1. Total 2.1M parameters.
- Imagenet-32: convolution layers have channels 3-64-128-256-512 with kernel size 3 and strides 1-1-2-1. Transposed convolution layers have channels 512-256-128-64-3 with kernel size 3-4-3-3 and strides 1-2-1-1. Total 3.3M parameters.

We remark that because inputs into the JKO-iFlow blocks are latent-space codes that have much lower dimensions than the original image, our JKO-iFlow blocks are simpler and lighter in design than models in previous NeuralODE [Grathwohl et al., 2019, Finlay et al., 2020] and diffusion model works [Ho et al., 2020, Song et al., 2021, Boffi and Vanden-Eijnden, 2023]. For instance, the DDPM model [Ho et al., 2020] on CIFAR10 has 35.7M parameters with more sophisticated model designs.

*Training specifics:* On MNIST, we fix the batch size to be 2000 during training, and we train 15K batches per JKO-iFlow block. We fix the learning rate to be 1e-3.

On CIFAR10 and Imagenet-32, we fix the batch size to be 512 during training, and we train 75K batches per JKO-iFlow block. The time per batch is 0.18 seconds on Imagenet-32 and 0.15 seconds on CIFA10. The total time on Imagenet-32 is 30 hours and on CIFAR10 is 24 hours. When training the blocks, the initial learning rate `lr0` is decreased by a constant factor of 0.9 every 2500 batches.

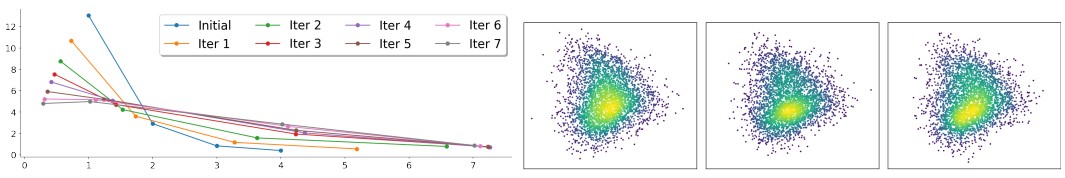

(a) Per-block $W_2^2$ over reparameterization iterations.   (b) Results at initial training (middle) and Iter 7 (right).

Figure A.5: Reparametrization iterations of JKO-iFlow model on MINIBOONE. (a) After 7 reparameterization iterations, a trajectory of more uniform $W_2$ movements is obtained. (b) Generated samples by the models before and after the moving iterations (visualized in the first two principal components computed from test data).

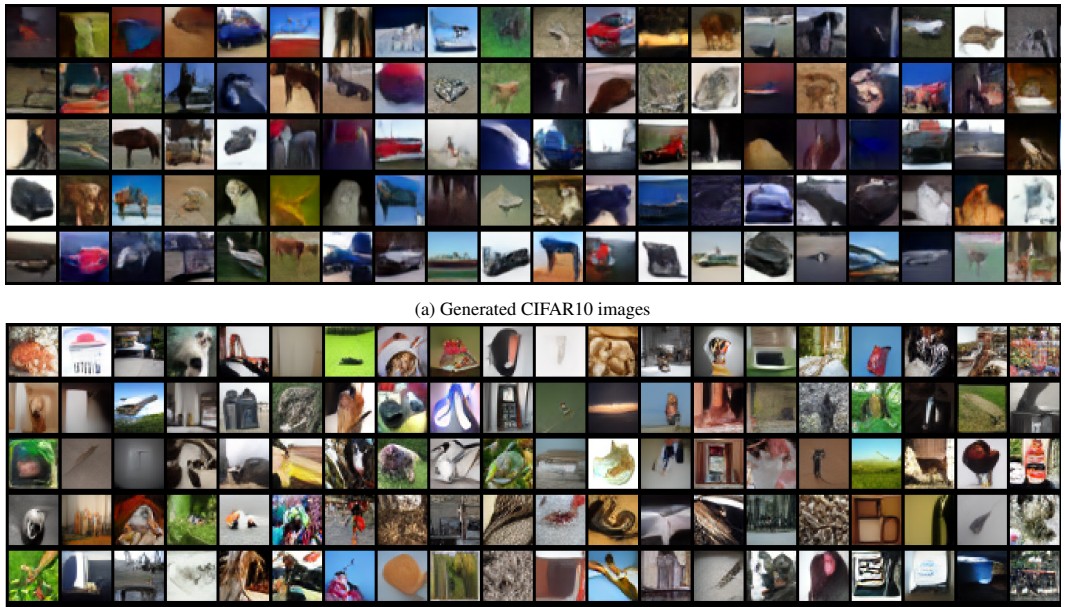

(a) Generated CIFAR10 images

(b) Generated Imagenet-32 images

Figure A.6: Uncurated generated samples of CIFAR10 and Imagenet-32 by JKO-iFlow in latent space.

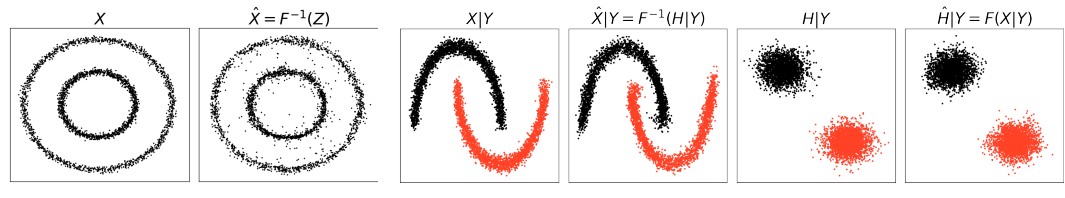

(a) Two-circles data (unconditional)      (b) Two-moons data (conditional)

Figure A.7: Unconditional and conditional generation on simulated toy datasets by JKO-iFlow. We color samples in (b) by the class label $Y$ taking binary values. In both (a) and (b), the generated samples are close to the true samples in distribution. In (b), the pushforward distribution by JKO-iFlow, denoted as $\hat{H}|Y$, is also close to the target Gaussian mixture distribution $H|Y$.

We let `lr0` be 1e-3 when training the first 5 blocks on Imagenet-32 and decrease it to be 8e-4 on blocks 6-8. We let `lr0` be 1e-3 when training the first 2 blocks on CIFAR10 and decrease it to be 7.5e-4 on the rest 6 blocks. Additionally, we use gradient clipping of max norm 1 after collecting gradients on each mini-batch.

*Additional results:* We perform the following steps to curate generated samples for CIFAR10 and Imagenet-32, which are shown in Figure 4, we first train an image classifier (i.e., VGG-16 [Simonyan and Zisserman, 2015]) on the training data. Then, we generate a large number of uncurated images (200K for Imagenet-32 and 20K for CIFAR10), classify them using the pre-trained classifier, and sort them based on top-1 predicted probability. Out of the top 20K images for Imagenet-32 and top 750 images for CIFAR10 upon sorting, we manually select images that we think most resemble the given predicted class. Figure A.6 further shows uncurated images on CIFAR10 and Imagenet-32 by the same JKO-iFlow model after training.

### C.2.4 Conditional generation

To modify the JKO-iFlow to apply to the conditional generation task, we follow the framework in Xu et al. [2022], which trains a single flow mapping from $X$ to $H$ that pushes to match each component of the distribution associated with a distinct output label value $Y = k$, $k = 1, \cdots, K$. Specifically, for a data-label pair $\{X_i, Y_i\}$, consider the continuous ODE trajectory $x(t)$ starting from $x(0) = X_i$, the per-sample training objective is by changing the term $V(x(t_{k+1}))$ in (8) to be $V_{Y_i}(x(t_{k+1}))$, where $V_k(\cdot)$ is the potential of the Gaussian mixture component $H|Y = k$. Because the Gaussian

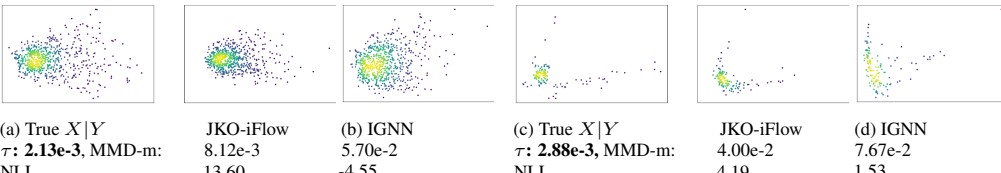

| (a) True $X|Y$ | JKO-iFlow | (b) IGNN | (c) True $X|Y$ | JKO-iFlow | (d) IGNN |
|---|---|---|---|---|---|
| $\tau$: **2.13e-3**, MMD-m: | 8.12e-3 | 5.70e-2 | $\tau$: **2.88e-3**, MMD-m: | 4.00e-2 | 7.67e-2 |
| NLL | -13.60 | -4.55 | NLL | -4.19 | 1.53 |

Figure A.8: Conditional graph nodal data generation by JKO-iFlow and iGNN on Solar ramping data. The nodal data $X \in \mathbb{R}^{10 \times 2}$ and the nodal label $Y \in \{0, 1\}^{10}$. Plots (a)(b) and (c)(d) are for two different sets of values $Y$ across all nodes respectively. Samples $X$ (concatenated across all nodes) are visualized by projection to two principal components (determined by true samples $X|Y$).

mixture is parametrized by mean vectors (the covariance is isotropic with fixed variance) [Xu et al., 2022], the expression of $V_k(\cdot)$ is a quadratic function with explicit expression.

*Training and test data:*

- For the simulated two-moon data, we re-sample 5K training samples every epoch, for a total of 40 epochs per block. The batch size is 1000.

- The solar dataset is retrieved from the National Solar Radiation Database (NSRDB), following the data pre-processing in [Xu et al., 2022]. The dataset has a 1K training samples and a 1K test samples. The batch size is 500, and each block is trained for 50 epochs.

*Choice of $h_k$:* For both datasets, $h_0 = 1, \rho = 1, h_{\max} = 3$. The reparametrization and refinement techniques are not used.

*MMD metric:* The MMD values are computed for each specific value of $Y$ (across all graph nodes). When $Y$ is fixed, we retrieve test samples from true data, denoted as $X|Y$, and generate model distribution samples, denoted as $\tilde{X}|Y$, and then compute the MMD values the same as in the unconditional case. Due to the relatively small sample size of $X|Y$ (679 and 144 respectively), we report the average values of MMD-m and threshold $\tau$ over 50 replicas. In each replica, we subsample 90% observations of $X|Y$ and generate $M = 2000$ samples to form $\tilde{X}|Y$. For the results in Figure A.8, for the $Y$ in plots (a)(b), $N = 612$ samples are randomly sampled from $X|Y$ in each replica, and the standard deviation of MMD-m values is less than the 2e-3 level. For the $Y$ in plots (c)(d), $N = 130$ samples are randomly sampled from $X|Y$ in each replica, and the standard deviation of MMD-m values is less than the 5e-3 level. The median distance kernel bandwidth computed on the entire $X|Y$ is used in computing the MMD values.

*Network and activation:* Regarding design of residual blocks,

- On two-moon data, we use fully-connected residual blocks with two hidden layers under 128 hidden nodes. The activation is Tanh. We train four residual blocks. The learning rate is 5e-3.

- On solar data, we follow the same design of residual blocks as [Xu et al., 2022]. More precisely, each residual block contains one Chebnet [Defferrard et al., 2016] layer with degree 3, followed by two fully-connected layers with 64 hidden nodes. The activation is ELU. We train 11 residual blocks. The learning rate is 5e-3.

