# OpenReview forum: "Normalizing flow neural networks by JKO scheme"
_NeurIPS.cc/2023/Conference — NeurIPS 2023 spotlight_

### Official Review · Reviewer_3zMK · 2023-06-29

**Soundness:** 4 excellent
**Presentation:** 3 good
**Contribution:** 4 excellent
**Rating:** 8
**Confidence:** 3

**Summary:**

The paper introduces a novel approach to train continuous normalizing flows/score-based diffusion models that exploits the Wasserstein gradient flow theory and the JKO iterative scheme.

The main idea is to approximate the density evolution of the variance-preserving forward dynamics of a diffusion model model (i.e. an Ornstein -Uhlenbeck process) with a series of deterministic transport maps. Under some conditions, the resulting deterministic dynamics is invertible and can then be used to map the stationary distribution back to the (implicitly learned) data distribution.

In practice, this is very similar to what is done in ordinary score-based diffusion models, but the networks are trained in a radically different way since with the JKO approach the score is trained indirectly through a deterministic algorithm.


**Strengths:**

I find this work exciting. The approach is very original as introduces a brand new way to train score-based models, which is backed by serious theoretical arguments. This both increases my confidence in the algorithm and opens the door for possibly very fruitful connection between several important research areas.

The paper is well-explained, although it relies on several complex concepts that could alienate some readers. However, I think that in this case the mathematical complexity is unavoidable and gives added value.

I appreciate that the authors not not solely focus on the mathematical framework but also discuss several important issues in the algorithmic implementation.

The experimental section is convincing and rather comprehensive.  The results on the low-dimensional datasets are good when compared with other state-of-the-art methods. The results on images are less impressive but overall convincing. While, in general, the application to generative computer vision needs to be fully flashed out, I agree with the authors that this goes beyond the scope of this work.


**Weaknesses:**

I think that, generally speaking, the paper does not have major weaknesses. However, the part of the experiment section dedicated to image generation would have benefited from having a larger scope, for example by including experiments on CelebA HD and ImageNet.

The selection of evaluation metrics should be complemented with FID scores, since this is the standard metric in the image generation literature and it is generally more reliable than the BPD score. Note that I do not think that having a lower FID score than the baseline is an argument for rejection, given the theoretical and algorithmic novelty of the work.

Ther exposition would benefit from a more extended section connecting the JKO approach to the standard score-matching diffusion theory. This would greatly help readers coming from the diffusion literature, who in my opinion are one of the most important audiences for this work.


**Questions:**

- Can you elaborate on the connections between the JKI iterative training and the score-matching training of diffusion models? It would be informative to discuss differences and similarities and give an overview of the strengths and weaknesses of each approach.

- I would like to see the FID scores on MNIST and CIFAR10.


**Limitations:**

Nothing to discuss.

---

> ### Author Rebuttal · Authors · 2023-08-09
>
> Please refer to the global response for the questions on experiments of image generation and FID scores.
>
> * **(Weakness 3rd paragraph) The exposition would benefit from a more extended section connecting the JKO approach to the standard score-matching diffusion theory. This would greatly help readers coming from the diffusion literature, who in my opinion are one of the most important audiences for this work.**
>
> We thank the reviewer for the suggestion. The connection between the JKO approach and the standard score-matching diffusion model is summarized below, and we will add it to the revised manuscript.
>
> First, it is known that the diffusion SDE for the stochastic process formed by the particles is connected to a deterministic ODE via the FPE (Fokker-Planck Equation). Specifically, the solution of the FPE Eqn (3) of the (variance preserving) SDE is equivalent to that of the continuity equation (CE) of the ODE Eqn (2) when the velocity field $f(x,t)$ is set to Eqn (9). While this relation has been used in the reverse-time (generating) process of score-matching diffusion model [Song et al., 2021], the JKO approach here is trying to directly learn a discrete-time ODE forward process and avoids sampling in training. The forward process approximates the CE and equivalently the FPE when step-size is small thanks to the JKO theory [Jordan et al., 1998].
>
> Meanwhile, unlike the score-based diffusion model, which tries to learn the score $\nabla \log \rho_t$, or equivalently the velocity field $f(x,t)$ that leads to exact solution of the FPE, with finite step size the JKO solution only approximates the solution of FPE. While such approximate sequence of densities $p_k$ do not have interpretation in physics, for the problem of normalizing flow it actually can provide a good CNF model without learning the exact solution of FPE: this is because as long as the last $p_k$ can be close enough to the normal density, and the transport in each learned residual block $T_k$ can be accurately inverted, this will guarantee the accuracy of the generating the data density in the reverse process. In this view, the JKO approach can explore the CNF models that transport from normal density to data density and back without requiring the intermediate process to exactly follow the diffusion process. The block-wise training scheme also allows savings and flexibility both in neural network architecture and in the training algorithm.
>
> Ref:
>
> Song et al. Score-Based Generative Modeling through Stochastic Differential Equations. ICLR. 2021.
>
> Jordan, Richard, David, Kinderlehrer, and Felix, Otto. "The Variational Formulation of the Fokker–Planck Equation". SIAM Journal on Mathematical Analysis 29, no.1 (1998): 1-17.
>
> * **(Question 1) Can you elaborate on the connections between the JKO iterative training and the score-matching training of diffusion models? It would be informative to discuss differences and similarities and give an overview of the strengths and weaknesses of each approach.**
>
> Thanks for the suggestion. We will explain the difference and similarities of the two in the “connection” section as in the answer to the previous question, and also add the following summary of strengths and weaknesses to the draft.
>
> The score-matching training has the advantage of more efficient optimization of the neural network that parametrizes the score function, thanks to the score-matching objective. In comparison, the current neural ODE model used in the JKO approach involves more expensive backpropagation of the neural network due to the ODE solver and the Hutchinson projection which undermines the scalability to higher dimensionality. Yet by paying this price, the CNF model has the advantage of optimizing the likelihood as the training objective and the ability to compute the likelihood by time integration.
>
> However, in training and generation the JKO approach has the advantage that it computes deterministic dynamics in both forward and reverse processes, avoiding simulating the SDE trajectories (injecting noise). The ODE integrator in JKO approach allows us to choose a much larger step size in the discrete time process compared to what is needed by the SDE simulation of score-based models.
>
> Another advantage of the JKO approach lies in the block-wise training. In addition to the savings of computation and memory, the block-wise training of the JKO-iflow model has the potential flexibility that different blocks may have different architectures, which may be advantageous in certain settings. This type of flexibility may not be easy to implement in the end-to-end training of most score matching diffusion models. Finally, the JKO scheme is general and may be combined with other flow-based backbone models to overcome the computational issue, possibly with specific designs for certain applications, see more in the answer to R1.

---

> > ### Comment · Reviewer_3zMK · 2023-08-14
> >
> > Thank you for the rdetailed response. I am happy keep my score and to recommends acceptance.

---

### Official Review · Reviewer_xTiV · 2023-07-03

**Soundness:** 3 good
**Presentation:** 2 fair
**Contribution:** 3 good
**Rating:** 7
**Confidence:** 3

**Summary:**

This paper presents an innovative method for enhancing the stability of trajectories in CNF (Continuous Normalizing Flow) models. The authors propose incorporating the JKO scheme, which introduces regularization between the current density and the base distribution. The core concept revolves around leveraging the Wasserstein distance to learn an optimal transport map that guides the data distribution towards a normal equilibrium. To assess the effectiveness of the model, experiments are conducted on various scenarios, including toy examples, tabular data, and datasets characterized by higher dimensions like CIFAR or MNIST.

**Strengths:**

- The idea behind the paper is novel and non-trivial.
- Optimizing the velocity field instead of the transport map using JKO-Scheme is intriguing.
- The procedure of training the model given the Algorithm is very efficient compared to reference baselines.
- The method that can be seen as a particular case of regularisation of CNF models achieves outstanding results on tabular benchmark datasets.
- The authors consider higher-dimension cases by conducting experiments on CIFAR. It is not common for such papers.


**Weaknesses:**

- The proposed method still does not provide good-quality image samples. This group of models is rather dedicated to obtaining high values of NLL and modeling the distributions of low-dimensional data.
- The variety of experiments conducted in this work may be richer. In particular, it would be interesting to see how the proposed approach works in some scenarios where CNFs perform well, like modeling multidimensional probabilistic regression, using the model in latent space of VAEs, or as a plugin model to large models (like StyleFlow).

**Questions:**

I do not have any particular questions for the authors.

**Limitations:**

The main limitation of the proposed approach is the lack of capabilities to generate good-quality of image data, but this group of methods is rather not designed for such problems. Moreover, the evaluation may be a bit broader and include some other applications mentioned in the "weaknesses" section. Besides that, I admire the contribution of the work. and I think it should be accepted to the conference.

---

> ### Author Rebuttal · Authors · 2023-08-09
>
> Please refer to the global response for the question on image quality and additional results of image generation using VAE latent space.
>
> * **(Weakness 2) The variety of experiments conducted in this work may be richer. In particular, it would be interesting to see how the proposed approach works in some scenarios where CNFs perform well, like modeling multidimensional probabilistic regression, using the model in latent space of VAEs, or as a plugin model to large models (like StyleFlow).**
>
> We thank the reviewer for suggesting a rich class of applications which are suitable scenarios to apply CNF models. For the latent space of VAEs, our real image generation experiments (CIFAR10 and the newly added Imagenet-32) are actually obtained by applying the JKO-iFlow model in the VAE latent space, see more in the global response above. We agree that other problems like multi-dimensional probabilistic regression [Chen et al., 2018] and plugins to deep architectures like StyleFlow [Abdal et al., 2021] are interesting extensions. We will add all this to the discussion of future directions.
>
> Ref.
>
> Chen R T Q, Rubanova Y, Bettencourt J, et al. Neural ordinary differential equations. NeurIPS 2018.
>
> Abdal R, Zhu P, Mitra N J, et al. Styleflow: Attribute-conditioned exploration of stylegan-generated images using conditional continuous normalizing flows. ACM Transactions on Graphics (ToG), 2021.

---

> > ### Comment · Reviewer_xTiV · 2023-08-16
> > **Thank you**
> >
> > Dear authors, thank you for the rebuttal. I maintain my decision, and I believe the paper should be accepted.

---

### Official Review · Reviewer_uXKQ · 2023-07-07

**Soundness:** 3 good
**Presentation:** 3 good
**Contribution:** 3 good
**Rating:** 7
**Confidence:** 2

**Summary:**

This paper proposes learning an invertible normalising flow using a neural ODE as a unique solution to the Fokker-Planck Equation (FPE) of the transport problem from the data distribution to the equilibrium solution. The solution of the FPE is obtained by using the JKO scheme by formulating it as a variational problem. The critical contribution of the paper is that each residual block of invertible neural ODE corresponds to a JKO step and the training objective can be computed from pushed data samples through the previous blocks. This leads to a block-wise procedure to train the JKO-iFlow model which is computationally more efficient than an end-to-end approach in the literature. The paper also proposes a way to determine the number of blocks adaptively by adaptively reparameterize the computed trajectory in the probability space with refinement to improve the model accuracy and the overall computational efficiency. Lastly, they empirical show that their method is competitive or better generative performance compared to existing flow and diffusion models on synthetic and real data,

**Strengths:**

This paper is novel, and the idea of using iterative block-wise training is a good one. This strategy can be applied to other works as well and I think is a good idea. The formulation of the loss function from the JKO-step is also very well-written and clear. One strength of the paper is that their proposed method is very flexible as they used neural ODE with sufficiently small step sizes in the residual blocks to invertibility making their method very general and doesn't have any restrictive architecture like is typical of neural of other invertible flow models.

**Weaknesses:**

The experiments are on relatively small datasets. It would be instructive to test how the model performs on larger datasets like miniImagNet or even CelebA dataset. The MMD loss could try to use different kernels or even mixed kernels to evaluate the quality better of the generative models. When comparing JKO-iflow against DDPM, it might be valuable to discuss what noise scheduling for DDPM is being used for each dataset, as DDPM is sensitive to the noise schedule used. Thus, it might lead to an unfair comparison if DDPM are not properly tuned. The authors can also include inverse problems in their experiments as many one of the advantages of DDPM is the fact they can easily be adapted to tackle inverse problems as well.

**Questions:**

Could there be probability trajectories that are particularly "stiff" much like how there are stiff ODE that require very small step sizes for the numerical ODE method to be stable and have the resulting solution be close to the actual trajectory?  The FPE does not only appear in image generation, in mean-field game theory the fokker-planck equation describes the dynamics of the aggregate distribution of agents can JKO-iflow work well on these other problems ?

**Limitations:**

The authors adequately address the limitations of their work, and there are no obvious potential negative social impacts of their work.

---

> ### Author Rebuttal · Authors · 2023-08-09
>
> Please refer to the global response for the questions on experiments of image generation.
>
> * **Different kernels in MMD to evaluate generative models**
>
> We agree with the reviewer that the MMD loss metric depends on the choice of the kernel, and a variety of kernels may be adopted for evaluating the trained generative model. In our experiments, we adopt the Gaussian kernel to stay consistent with reported baselines from the literature [Onken et al., 2021] for the purpose of comparison. Being aware of the sensitivity of the kernel MMD to the choice of the kernel parameter, we performed MMD calculations with different bandwidth selections. In addition, to ensure that the comparison of MMD values is meaningful, we applied bootstrap to compute the threshold of the MMD test under null (when two distributions are the same), and all values to be compared are above the threshold. The details of the MMD tests, including bandwidth selection, are provided in Appendix C.1.2. Further evaluation of the generative models using more metrics would be an interesting direction for future work.
>
> * **Noise schedule for DDPM to ensure fair comparison**
>
> We thank the reviewer for the suggestion, and have conducted additional experiments of different noise schedulers following DDPM [Ho et al., 2020] to ensure fair comparison. Below, we use three schedulers (i.e., the $\beta(t)$ definition in [Song et al., 2021]) with five different choices for $\bar{\beta}_{\max}$ in the variance-preserving ScoreSDE model, for which DDPM can be viewed as a discrete-time version. As shown in Table 1 below, the performance of DDPM is indeed sensitive to the noise schedule. However, the best NLL of DDPM 17.45 from the table is still noticeably higher than the NLL of 12.55 obtained by JKO-iFlow, and the latter is trained using ten times less number of batches.
>
> Table 1: NLL per noise scheduler and $\bar{\beta}_{\max}$ combination, the three settings ‘linear, constant, quadratic’ follow the DDPM paper [Ho et al., 2020]. The mean and standard deviation are computed over three replicas of the trained model
> |Noise Scheduler\\ $\bar{\beta}_{\max}$|1|5|10|15|20|
> |-|-|-|-|-|-|
> |linear|24.51 (0.24)|18.73 (0.64)|20.28 (0.38)|26.76 (1.77)|26.83 (1.23)|
> |constant|25.47 (0.25)|30.37 (0.84)|37.73 (0.98)|40.47 (1.27)|45.32 (0.40)|
> |quadratic|27.19 (0.14)|17.45 (0.17)|18.48 (0.38)|18.90 (0.48)|21.35 (0.60)|
>
> Ref.
>
> Ho et al. Denoising diffusion probabilistic models. NeurIPS 2020
>
> Song et al. Score-based generative modeling through stochastic differential equations. ICLR 2021.
>
> * **To include inverse problems in experiments for which DDPM has advantage**
>
> We appreciate the reviewer's suggestion on the application to inverse problems. For the application to image data, image restoration problems can be cast as a conditional generation task. In this work, we have investigated conditional generation in Section 5.5 exemplified by graph data, where we applied the proposed model to learn from data the conditional distribution $X|Y$, where $X$ and $Y$ are nodal descriptors and labels on a graph, respectively. We demonstrated that JKO-iFlow outperforms the previous baseline in terms of generative quality, NLL, and MMD metrics. This setting may also be viewed as an inverse problem. The extension to other inverse problems is an interesting future direction.
>
> * **Can probability trajectories be stiff and require small step size to ensure stability and accuracy of the numerical ODE?**
>
> We thank the reviewer for asking the interesting question. First, in the context of JKO-iFlow, a neural ODE block is used to parametrize the $k$-th step transport map $T_k$, which pushwards from $p_{k-1}$ to $p_k$. Theoretically, as long as both distributions have smooth densities, there exists a smooth transport map in between [Villani 2021]. Meanwhile, since the JKO scheme in the case of a small step size approximates the solution of a diffusion process FPE, which smoothes the distributions as time increases, we expect the discrete-time densities $p_k$ to also become more and more smooth as $k$ increases. In the case that the initial-time density (the data distribution) is not regular, this will make the flow at the early time irregular, and we observed that this is the place where “stiffness” may more often present. As the reviewer pointed out, this can require a small step size of the neural ODE.
>
> The time-reparametrization introduced in Section 4.2 can be viewed as an adaptive way to adjust the step size to overcome this difficulty, and we have shown that this improves the performance of the JKO-iFlow model in experiments, specifically on toy data and MINIBOONE tabular data (see Sections 5.2 and 5.3). There are other works that address ODE stiffness and improve the stability of neural ODE, such as [Kim et al., 2021], which can be readily incorporated into our framework if needed.
>
> Ref.
>
> C´edric Villani. Topics in optimal transportation, volume 58. American Mathematical Soc., 2021.
>
> Kim S, Ji W, Deng S, et al. Stiff neural ordinary differential equations. Chaos: An Interdisciplinary Journal of Nonlinear Science, 2021, 31(9).
>
> * **FPE also appears in mean-field games, and can JKO-ifow work well on these other problems?**
>
> We thank the reviewer for suggesting the application of the JKO-iFlow model to broader problems of mean-field games (MFG). The current work focuses on the continuous normalizing flow (CNF) setting and the application to generative models. Under the framework of MFG from time $0$ to $T$, if one sets the terminal cost as the $KL$-divergence between the density $\rho_T$ and $p_Z = N(0, I)$, then the CNF problem is a special case of the MFG problem. We think that the JKO-iFlow model potentially can be applied to other MFG problems, and such extensions would be a very interesting future direction.

---

### Official Review · Reviewer_NZGt · 2023-07-09

**Soundness:** 3 good
**Presentation:** 3 good
**Contribution:** 3 good
**Rating:** 7
**Confidence:** 3

**Summary:**

The authors introduce a novel normalizing flow training algorithm that integrates continuous normalizing flows and the JKO scheme. The objective of the training algorithm is to minimize KL divergence between the current density and the equilibrium density. The proposed method is inspired by the JKO scheme to enable adaptive blockwise training of residual networks. Furthermore, the authors demonstrate their method on synthetic and open-source datasets.

**Strengths:**

1). The proposed method is computationally efficient compared to other normalizing flow methods or diffusion.

2). Novel idea of optimizing the velocity field instead of the transport map motivated JKO-Scheme.

3). The paper is well-motivated and easy to understand.

**Weaknesses:**

1). The paper lacks metrics/empirical validation on larger images. FID metrics for Figure 4 should be necessary considering the paper is about generative modeling and its challenging to tell how adequate the CIFAR-10 samples are because they have such low resolution via eye test.

2). A significant highlight is that block-wise training leads to a large reduction in computation cost compared to other popular generative models (Figure 4), but the authors do not highlight that the proposed scheme uses a pre-trained auto-encoder, where some of the other methods do not.



**Questions:**

1). Is it possible for the authors to provide FID scores for both MNIST and CIFAR-10?

2). How good is the proposed method without using a pre-trained auto-encoder?



**Limitations:**

The limitations are clearly discussed.

---

> ### Author Rebuttal · Authors · 2023-08-09
>
> Please refer to the global response for the questions on experiments of image generation and FID scores.
>
> * **(Weakness 2 & Question 2): The proposed scheme uses a pre-trained auto-encoder, where some of the other methods do not. How good is the proposed method without using a pre-trained auto-encoder?**
>
> We thank the reviewer for asking the good question. We mainly propose to use the current method in the latent space for image generation tasks, which will be clarified in the revised draft. Directly applying the current approach to image pixel space is possible, but would incur high computational cost due to the likelihood-based training of the neural ODE backbone model, see more in the global response on "the extension to larger images". Consequently, the computational advantage of the block-wise training would be discounted.
>
> At the same time, we would like to point out that the block-wise training of the JKO scheme aims to compute a sequence of transported distributions that get closer to the normal density progressively, and this may be combined with other backbone models that are more suitable for computation in image pixel space. One possibility is by adopting the flow models trained to match the desired velocity field [Albergo et al., 2023; Lipman et al., 2023]. We will clarify the limitations and discuss future directions in the discussion section.
>
> Ref.
>
> Albergo M S, Vanden-Eijnden E. Building normalizing flows with stochastic interpolants. ICLR 2023.
>
> Lipman Y, Chen R T Q, Ben-Hamu H, et al. Flow matching for generative modeling. ICLR 2023.

---

> > ### Comment · Reviewer_NZGt · 2023-08-14
> > **Thank you for the response**
> >
> > I want to thank the authors for their detailed responses. I am happy to keep my score the same and recommend this paper for acceptance.

---

### Author Rebuttal · Authors · 2023-08-09

Thanks for the constructive comments and feedback provided by all the reviewers. The common questions are about the experiments on image generation, which we first address here. The additional questions and comments of each reviewer are addressed in the specific responses below. R1 = Reviewer NZGt, R2 = Reviewer uXKQ, R3 = Reviewer xTiV, R4 = Reviewer 3zMK.

* **Quality of generated images, quantitative evaluation (FID), and additional image examples. [R1 Question 1, R3 Weakness 1 & Limitations, R4 Question 2]**

We have conducted additional experiments on CIFAR10. Please see the attached PDF for generated images by the final model, which show improved image quality. The model achieves an FID of 29.10. The improvement from the initial submission comes from using a larger VAE model following [Rombach et al., 2022] and longer training time (24 hours on one A100 GPU, the VAE training is extra). We have also computed the FID for MNIST, which is 7.95 using JKO-iFlow in the code space of a pre-trained VAE.

To enlarge the image generation experiments, we further examined our model on ImageNet-32, which consists of 1.28 million training images. The generated images are also shown in the attached PDF, and the model achieves an FID of 20.10 after training for 30 hours on one A100 GPU. We are aware that these FIDs do not achieve the SOTA performance of some diffusion-based models, but these results are obtained with less computation and this image generation performance is better than most CNF baselines. We will incorporate these results in the paper revision.

* **Extension to larger images [R1 Weakness 1, R2 Weakness (1st-2nd sentences), R4 Weakness paragraphs 1 & 2]**

We thank the reviewers for the suggestion, and larger images are certainly an important application direction of the proposed approach. Our current method for image generation is by applying a flow model in the latent VAE space. Note that generative models in the latent space, like StableDiffusion [Rombach et al., 2022], obtain SOTA image generation and are popular approaches for many industry applications. Thus, the VAE+flow approach has the potential to generate larger images.

Meanwhile, we do think the extension to larger images would be more efficient if some fundamental development of the neural ODE can be done in the first place. A main reason is that the neural ODE backbone model currently taken by this work does not have a strong computational scalability to high-dimensional space due to the expensive backpropagation via ODE solver and the Hutchinson projection to compute $div(f)$ in the likelihood-based training. The current JKO approach is for general data and not specifically designed for large image generation tasks (as has been pointed out by R3). To extend the JKO approach to larger images, one can proceed by developing a more efficient neural ODE to scale to higher dimensional latent or pixel space or by combining the JKO scheme with other backbone models more suitable for image generation tasks, see more in the answer to R1 below. We leave these developments to future work and will add all this to the discussion.

Ref.

Rombach R, Blattmann A, Lorenz D, et al. High-resolution image synthesis with latent diffusion model. CVPR. 2022

---

### Decision · Program_Chairs · 2023-09-21

**Decision:**

Accept (spotlight)

**Comment:**

In this paper, the authors introduce a neural ODE based formulation of a normalizing flow generative model.  The architecture of the formulation is composed of residual blocks that correspond to JKO Steps from a particular transport problem.  The authors provide a block-wise procedure to train this model and justify that it is more computationally efficient than end-to-end approaches in the literature.  The resulting approach is shown to have competitive or better performance compared to existing flow and diffusion models on image datasets that are small in image size.  The papers strengths include the signifiant novelty and motivation of a JKO inspired normalizing flow formulation, the computational efficiency of the proposed approach, and its competitive performance relative to related methods.  Reviewers noted that the method was tested on relatively small images, and the paper could be improved by evaluation of images of a larger size.  Given the difficulty of conducting such experiments on larger images, the strengths of the paper outweigh this drawback.